# AutoAgent: A Fully-Automated and Zero-Code Framework for LLM Agents

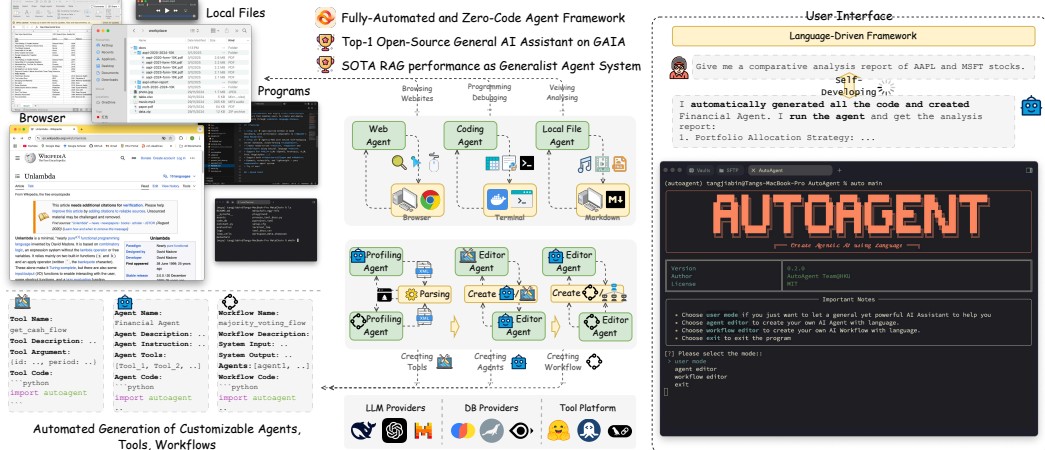

Figure 1: AutoAgent stands out as a new LLM Agent Framework that enables fully automated, zero-code development for complex task automation. Ranking #1 among open-source solutions on the GAIA benchmark, it delivers state-of-the-art RAG performance as a general AI assistant. Its revolutionary approach democratizes AI development - allowing anyone, regardless of coding experience, to create and customize their own agents, tools, and workflows with ease.

## ABSTRACT

Large Language Model (LLM) Agents have demonstrated remarkable capabilities in task automation and intelligent decision-making, driving the widespread adoption of agent development frameworks such as LangChain and AutoGen. However, these frameworks predominantly serve developers with extensive technical expertise—a significant limitation considering that only 0.03% of the global population possesses the necessary programming skills. This stark accessibility gap raises a fundamental question: *Can we enable everyone, regardless of technical background, to build their own LLM agents using natural language alone?* To address this challenge, we introduce AutoAgent - a **Fully-Automated** and highly **Self-Developing** framework that enables users to create and deploy LLM agents through **Natural Language Alone**. Operating as an autonomous Agent Operating System, AutoAgent comprises four key components: i) Agentic System Utilities, ii) LLM-powered Actionable Engine, iii) Self-Managing File System, and iv) Self-Play Agent Customization module. This lightweight yet powerful system enables efficient and dynamic creation and modification of tools, agents, and workflows without coding requirements or manual intervention. Beyond its code-free agent development capabilities, AutoAgent also serves as a versatile multi-agent system for **General AI Assistants**. Comprehensive evaluations on the GAIA benchmark demonstrate AutoAgent's effectiveness in generalist multi-agent tasks, surpassing existing state-of-the-art methods. Furthermore, AutoAgent's Retrieval-Augmented Generation (RAG)-related capabilities have shown consistently superior performance compared to many alternative LLM-based solutions. Code link: https://anonymous.4open.science/r/AutoAgent-NIPS-D4AF/.

# 1 INTRODUCTION

The emergence of Large Language Models (LLMs) has revolutionized AI agent development, enabling unprecedented breakthroughs in autonomous task execution and intelligent problem-solving. LLM-powered agents excel at understanding context, making informed decisions, and seamlessly integrating with various tools and APIs. Leading frameworks like LangChain LangChain (2023), AutoGPT Significant-Gravitas (2023), AutoGen Wu et al. (2023), CAMEL Li et al. (2023), and MetaGPT Hong et al. (2024a) have demonstrated remarkable success in automating increasingly complex workflows - from sophisticated web navigation to advanced data analysis and innovative creative content production. By leveraging advanced mechanisms such as role-playing, structured operating procedures, and dynamic agent coordination, these frameworks deliver exceptional problem-solving capabilities while significantly reducing human intervention.

Despite remarkable advancements in AI agent development, a significant barrier persists: the creation and optimization of LLM agent systems remains dependent on traditional programming expertise. Current frameworks primarily cater to technically proficient developers who can navigate complex codebases, understand API integrations, and implement sophisticated prompt engineering patterns. This reliance on coding skills creates a substantial accessibility gap, as only 0.03% of the global population possesses the necessary programming expertise to effectively build and customize these agents. Even with well-documented frameworks and development tools, the entry barrier remains dauntingly high for non-technical users. This limitation becomes particularly problematic given the universal need for personalized AI assistants in digital age. Everyone, from business professionals seeking workflow automation to educators designing interactive learning tools, requires customized LLM agents tailored to their specific needs. For instance, a researcher might need an agent specialized in literature review and data analysis, while a content creator might require an agent focused on creative writing and media management. The current paradigm of coding-dependent agent development not only severely restricts the user base but also creates a bottleneck in meeting the diverse and evolving demands for personalized AI assistance. This misalignment between universal needs and limited accessibility calls for a fundamental rethinking of how LLM agents are created and customized.

This stark contrast between universal needs and limited accessibility leads us to a fundamental research question: *Is it possible to democratize LLM agent development by enabling Natural Language-based Creation and Customization?* In this work, we aim to realize this vision by introducing AutoAgent, a novel framework that fundamentally reimagines agent development as a fully automated, language-driven process requiring zero programming expertise. To realize this vision, AutoAgentoperates as an autonomous Agent Operating System with three key capabilities: 1) **Natural Language-Driven Multi-Agent Building** - automatically constructing and orchestrating collaborative agent systems purely through natural dialogue, eliminating the need for manual coding or technical configuration; 2) **Self-Managing Workflow Generation** - dynamically creating, optimizing and adapting agent workflows based on high-level task descriptions, even when users cannot fully specify implementation details; and 3) **Intelligent Resource Orchestration** - providing unified access to tools, APIs, and computational resources via natural language while automatically managing resource allocation and optimization. Through this innovative architecture, AutoAgentdemocratizes LLM agent development while maintaining enterprise-grade sophistication, transforming a traditionally complex engineering task into an intuitive conversation accessible to all users.

To enable fully-automated and zero-code LLM agent development, AutoAgent introduces several synergistic technical innovations that form a complete framework: First, the **Agentic System Utilities** provides a foundational multi-agent architecture, where specialized web, code, and file agents collaborate seamlessly to handle diverse real-world tasks. At its core, the **LLM-powered Actionable Engine** serves as the system's brain, supporting flexible integration of any LLM provider through both direct and transformed tool-use paradigms for robust action generation. To address the critical challenge of information management, the **Self-Managing File System** enhances overall system capability by automatically converting diverse data formats into queryable vector databases, enabling efficient information access across all operations. Additionally, the **Self-Play Agent Customization** not only transforms natural language requirements into executable agents through structured XML schemas, but also automatically generates optimized workflows through iterative self-improvement, eliminating the need for manual agent programming or workflow design. Together, these innovations enable AutoAgent to democratize agent development while maintaining production-level robustness.

AutoAgent's exceptional capabilities have been rigorously validated through comprehensive empirical evaluation. In standardized benchmarks, it secured a strong second place on the Generalist Agent Benchmark (GAIA), while significantly outperforming state-of-the-art RAG approaches on the Retrieval-Augmented Generation benchmark. Beyond these quantitative achievements, extensive case studies demonstrated AutoAgent's robust self-development capabilities across diverse real-world scenarios, highlighting its practical value in automated agent development.

## 2 RELATED WORK AND PRELIMINARIES

LLM-empowered agents have revolutionized AI systems through tool invocation capabilities. Frameworks like LangChain LangChain (2023), AutoGPT Significant-Gravitas (2023), CAMEL Li et al. (2023), MetaGPT Hong et al. (2024a), and OpenAgent Xie et al. (2023) demonstrate diverse capabilities: CAMEL pioneered role-playing communication, AutoGen Wu et al. (2023) developed LLM combinations for problem-solving, MetaGPT Hong et al. (2024a) integrated Standardized Operating Procedures, and OpenAgent Xie et al. (2023) provided specialized web agents. However, these frameworks require coding skills and domain expertise, limiting accessibility to non-technical users. We propose a new paradigm that democratizes agent development through natural language interactions, making agent technology accessible regardless of technical background.

**LLM-Empowered Agent**. The task-solving process of Large Language Model (LLM) agents can be formalized as a Markov Decision Process (MDP), providing a comprehensive framework for modeling their interaction with the environment. Defined as $\mathcal{M} = (\mathcal{S}, \mathcal{A}, O, P(\cdot), \mathcal{E})$, the MDP captures the agent's state space $\mathcal{S}$, action space $\mathcal{A}$, observation space $O$, state transition function $P(\cdot)$, and the set of environments $\mathcal{E}$ it can interact with. At each time step, the LLM agent observes the current state, selects an action based on its policy, interacts with the environment, and updates its state, often referred to as the agent's "context". The mapping from state to action can follow two primary paradigms: **Tool-Use** Yao et al. (2024), where the agent utilizes external capabilities, and **ReAct** Yao et al. (2023) (Non-tool-use), where the agent generates the next action solely based on its internal language model. This MDP formulation provides a powerful framework for understanding, analyzing, and designing LLM-empowered agents capable of tackling a wide range of complex, multi-step tasks.

**Generalist Multi-Agent System**. Multi-agent systems (MAS) overcome single agent limitations in complex problems by leveraging diverse capabilities of multiple specialized agents. Generalist Multi-Agent Systems employ agent teams coordinated by an Orchestrator to solve varied tasks through collaborative intelligence.

In a Generalist MAS, there are multiple agents, denoted as $\pi_0 : S_0 \to A_0, \pi_1 : S_1 \to A_1, \ldots, \pi_n : S_n \to A_n$. Within each agent's action set, there exists a special **transfer action** $\hat{A}_i \in A_i$, which enables the delegation of tasks to other agents. The key challenge in a MAS lies in designing an effective **Task Transfer Mechanism**, which organizes different agents through appropriate transfer actions. We define such an agent organization mechanism as the "MAS Design Pattern". A common design is the **Orchestrator-Workers** paradigm Fourney et al. (2024); Anthropic (2024), where the **Orchestrator** comprehends the task and distributes subtasks to **Workers** via transfer actions. The Workers, acting as sub-agents, execute the subtasks and return the results to the Orchestrator through transfer actions.

For tasks with deterministic steps, workflow-driven mechanisms have emerged as a particularly effective approach. Notable examples include GPTSwarm Zhuge et al., which models workflows as computational graphs for complex data operations, and specialized systems for mathematical reasoning Wang et al. (2023) and code generation Wang et al. (2024a). These advances have enabled strong capabilities across various applications, including GUI interaction Hong et al. (2024b); Xu et al. (2024), software development Wang et al. (2024b); Yang et al. (2024), web browsing Drouin et al. (2024); Song et al. (2024), and embodied tasks Li et al. (2024), demonstrating the versatility and potential of LLM-powered agent systems.

**Workflow Design in Generalist MAS**. While Generalist Multi-Agent Systems offer high flexibility, particularly for open-ended and complex tasks, they also come with higher costs, increased complexity, and potential for compounding errors Anthropic (2024). For tasks with deterministic processes and expert domain knowledge, fixed workflows can be a more stable and effective approach.

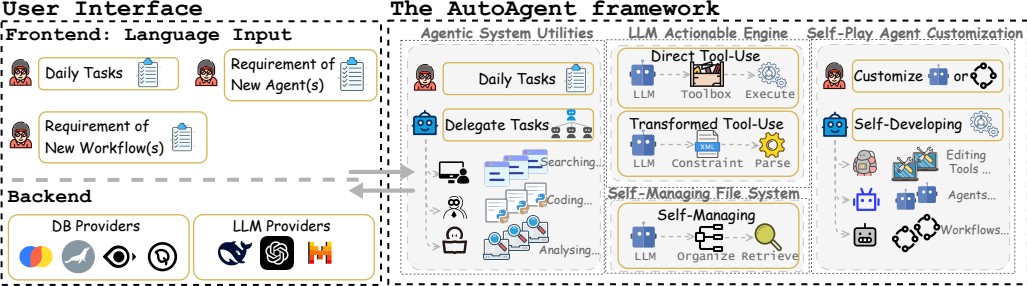

Figure 2: AutoAgent is a fully automated, language-driven generalist agent system. The core components that enable this include the Agentic System Utilities, the LLM-powered Actionable Engine, the Self-Managing File System, and the Self-Play Agent Customization module.

A workflow in a MAS is defined as $\mathcal{W} = w_{i,j} : \pi_i \xrightarrow{c_k} \pi_j$, where $\pi_i$ and $\pi_j$ represent agents, and $c_k$ is the transfer condition. Effective workflow design involves defining conditional transfer equations between agents based on downstream tasks, ensuring smooth and coordinated transmission - the "Workflow Design Pattern". Common patterns include **Routing** (directing tasks through a sequence), **Parallelization** (distributing subtasks concurrently), and **Evaluator-Optimizer** (using agents to assess and refine) Anthropic (2024). These can create stable and efficient task-solving in MAS.

**Fully-Automated Generalist MAS**. While the successful design and execution of both multi-agent systems and complex workflows typically require substantial expert knowledge and engineering expertise, the goal of AutoAgent is to fully automate this process. The key challenge lies in seamlessly bridging the gap from high-level user requirements to the practical implementation of effective MAS and workflow solutions - all through natural language interactions.

## 3 THE AUTOAGENT FRAMEWORK

AutoAgent is designed to be the automated operating system for LLM agents and general AI assistant. Inspired by modern computer operating systems, AutoAgent consists of key components that enable seamless natural language-driven agent development and task execution, as illustrated in Fig 2. Its **Agentic System Utilities** provide foundational building blocks for complex agent-driven tasks, while the **LLM-powered Actionable Engine** forms the central brain, understanding inputs and orchestrating multi-agent coordination. The **Self-Managing File System** manages structured storage and retrieval of user multi-modal data, and the **Self-Play Agent Customization** empowers users to generate specialized, tailored agents and workflows through natural language, without any coding requirements. Collectively, these robust capabilities make AutoAgent a versatile and powerful platform, powering a variety of autonomous agent-based solutions for diverse applications.

### 3.1 AGENTIC SYSTEM UTILITIES

The AutoAgent framework employs a modular, multi-agent architecture to address the key challenge of developing intelligent personal assistant agents capable of seamlessly integrating and coordinating diverse capabilities, from web browsing and information retrieval to data analysis and code execution. This design choice, which comprises specialized agents for web, coding, and file management tasks, as well as an orchestrator agent to decompose and delegate user requests, enables the agentic system utilities to serve as a versatile and extensible foundation that can adapt to a wide range of user requirements, facilitating the rapid development of tailored, agent-driven solutions. Detailed system prompts and tool definitions for **Agentic System Utilities** can be found in Appendix Sec A.

#### 3.1.1 ORCHESTRATOR AGENT

The Orchestrator Agent is the primary interface for interacting with the user. It receives tasks from the user, comprehends the tasks, decomposes them into sub-tasks, and delegates these sub-tasks to appropriate sub-agents using the `handoff` tools OpenAI (2024). Once a sub-agent completes a sub-task, it returns the result to the Orchestrator also using the `handoff` tool. Based on the task completion status, the Orchestrator continues to assign the next sub-task to a suitable agent. This iterative process continues until the entire task is completed. The Orchestrator, designed with the `handoff` mechanism, is a simple yet effective solution, eliminating the need for complex prompts to handle task planning.

#### 3.1.2 DEDICATED AGENT MODULE

The basic Agent system can be broadly categorized into three types: **web interaction, code execution, and file analysis.** We have constructed three functionally complementary dedicated agent modules: Web Agent, Coding Agent, and Local File Agent, each designed for these respective tasks.

**The Web Agent provides a flexible and extensible web-toolkit**, enabling the agent to perform a variety of web-based tasks, from general web searches to file downloads. Its core functionalities include web searching, page navigation, content browsing, and file downloading, which are abstracted into 10 high-level operational interfaces (e.g., `click`, `web_search`, `visit_url`, etc.). This module is built on BrowserGym Drouin et al. (2024), creating a browser environment that combines low-level code-driven actions to implement high-level tools, thereby enhancing the extensibility of tool definitions.

**The Coding Agent is a comprehensive solution for code-related tasks**, capable of handling a wide range of applications—from data processing and numerical computation to machine learning and system management. It offers diverse tools and supports execution in a secure, isolated interactive terminal environment. All code execution results are fed back to the agent via terminal output, with pagination support (e.g., using commands like `terminal_page_up`, `terminal_page_down`, and `terminal_page_to`. ), effectively bypassing the context length limitations of LLMs. To ensure security, all operations run within a Docker sandbox, with optional integration of third-party secure execution platforms like E2B E2B (2024).

**The Local File Agent is designed to uniformly process various local multimodal file types** (e.g., text documents, audio, video, spreadsheets) and convert them into structured formats for analysis. This agent standardizes different file formats (e.g., `.pdf`, `.mp4`, `.csv`, `.docx`) and displays them in a terminal-like Markdown browser with pagination, enabling efficient handling of long texts or complex file structures for everyday tasks.

These three agent modules work collaboratively, forming light-weight yet complete basic Agent System, laying the foundation for future functional expansion.

### 3.2 LLM-POWERED ACTIONABLE ENGINE

As the CPU executes instructions, manages resources, and coordinates processes in an OS, the LLM-powered actionable engine can understand natural language, generate plans, and coordinate tasks across agents. This enables seamless human-agent collaboration and task completion.

We utilize LiteLLM BerriAI (2024) to standardize LLM requests through an OpenAI-like interface, supporting 100+ models from various providers. For agent collaboration, the LLM receives all action-observation pairs up to time $t$ as state $s_t$ to determine the next action. These pairs serve as system RAM, facilitating efficient retrieval and enabling language-driven system coordination.

#### 3.2.1 GENERATING ACTIONABLE REFLECTIONS

We generate reflections (*i.e.*, actions) based on LLM context, which can be broadly categorized into two distinct approaches that leverage the language model's capabilities.

**Direct Tool-Use Paradigm**. This approach is suitable for commercial LLMs or LLM serving platforms that support tool-use. These LLMs can directly generate a parsed next-step tool to execute based on the provided tool set and the current state, reducing errors during the tool parsing phase. However, this method heavily relies on the optimization of the third-party platform's capabilities.

**Transformed Tool-Use Paradigm**. This approach does not rely on the LLM's inherent tool-use capabilities. Leveraging the superior code-generation abilities of modern LLMs, we transform the tool-use paradigm into a structured XML code generation task, *e.g.*, `<function=function_name>` `<parameter=parameter_1>value_1</parameter> ... </function>`. This structured output is then parsed to extract critical information like tool arguments and others. It improves the performance of commercial models with suboptimal tool-use capabilities and enables the integration of open-source LLMs into the system, providing greater flexibility and customization.

### 3.3 SELF-MANAGING FILE SYSTEM

The file system in AutoAgent is a vector database that LLM agents can retrieve and understand. In our design framework, users can upload text files in any format (e.g., `.pdf`, `.doc`, `.txt`) or compressed archives and folders containing any text files. The system tools in the file system automatically convert these files into a consistent text format and store them in a user-defined collection within the vector database (using the `save_raw_docs_to_vector_db` tool). This enables agents to self-manage their database memory and perform efficient and accurate retrieval and generation using tools like `query_db` and `answer_query`. The detailed definitions of the tools are presented in Tab 4.

### 3.4 SELF-PLAY AGENT CUSTOMIZATION

To allow users to customize tools and agents for specific scenarios or build their own multi-agent systems and workflows, it is designed as a code-driven, controllable self-programming agent framework. By implementing constraints, error-handling mechanisms, and customized workflows, it enables controlled code generation, facilitating the creation of tools, agents, and workflows. The AutoAgent supports two distinct modes: agent creation without workflow and agent creation with workflow.

#### 3.4.1 AGENT CREATION WITHOUT WORKFLOW

Building effective multi-agent systems often requires domain-specific expertise, such as in-depth knowledge of financial regulations or healthcare protocols. However, this level of specialized know-how may not always be available to users. For example, in the financial services, constructing a multi-agent system to automate complex investment portfolio management would necessitate expertise in areas like asset allocation, risk modeling, and regulatory compliance.

To address this challenge, our AutoAgent provides a powerful workflow-based mode allowing users to generate sophisticated agent systems with minimal domain expertise. In this mode, the user provides high-level descriptions of the desired agent(s), such as the agent's name and a brief sentence-level description for the expected agent functionalities. AutoAgent then uses this input to automatically generate the appropriate agent(s) and the necessary workflow(s) to orchestrate their collaborative efforts. This is all done based on the current state of the framework, including the available tools, agents, and workflows. The key steps in this workflow-based agent generation approach are:

•**Analyze Requirements and Existing Components**. The process begins by carefully analyzing the user's requirements in the context of the existing tools and agents already available in the system. This in-depth analysis is performed using the specialized profiling agent, which helps thoroughly assess the current capabilities and resources that can be leveraged to fulfill the user's needs.

•**Analyze Tools and Structure Agents**. Based on the comprehensive analysis performed, the system then carefully determines the need for creating new tools, meticulously evaluates whether existing tools can be effectively utilized, and subsequently structures the seamless collaboration between multiple agents as appropriate. This ensures the optimal and comprehensive use of available resources, ultimately leading to the efficient agent system design.

•**Generate Detailed XML Agent Specifications**. This step is designed to generate structured XML code that accurately represents the detailed agent creation requirements. This comprehensive XML representation captures the necessary information, including the agent's functionality, dependencies, and interactions, to enable the seamless and efficient subsequent process of agent generation.

**Optimized Tool Creation with Third-Party APIs**. The Tool Editor Agent can seamlessly integrate various third-party APIs, such as LangChain, RapidAPI, and Hugging Face, to create powerful tools. It expertly utilizes advanced retrieval techniques to search for and surface relevant API documentation, including comprehensive details like names, descriptions, and invocation methods. The robust system currently supports an extensive collection of 145 APIs from 8 diverse categories in RapidAPI, LangChain LangChain (2023), and a wide range of models from 9 categories in Hugging Face. Future plans include seamlessly integrating more cutting-edge platforms like Composio Composio (2024).

The agent also generates tool code based on its knowledge, automatically checking for syntax errors. It designs test cases, runs the tool, and verifies functionality. If the tool fails, the agent automatically debugs the code until successful. This targeted approach allows a more customized and adaptable tool set, rather than a bloated, integrated system.

**Agent Creation and Execution**. When the user's requirements involve multiple agents focused on different tasks, the Agent Editor Agent automatically identifies this need and performs the necessary multi-step agent creation operations. After all agents are successfully created, the system invokes the `create_orchestrator_agent` tool to generate an orchestrator agent that connects the required agents. This orchestrator adheres to the Orchestrator-Workers MAS design pattern, with a system prompt that includes task descriptions, sub-task decomposition rules, and other scenario-specific details. Detailed algorithms and system prompts are provided in Appendix Sec A.6.1.

#### 3.4.2 AGENT CREATION WITH WORKFLOW

When users have specific requirements for a MAS's workflow and domain knowledge, AutoAgent allows a tailored approach. In this mode, users provide descriptions of the desired agent(s) and specify the tasks they want the created agent(s) or workflows to accomplish. AutoAgent then uses this

information about the target tasks to generate not just the individual agent(s), but also the necessary workflow(s) to coordinate their collaborative efforts in achieving the specified objectives.

Traditional graph-based methods often require strict adherence to graph theory principles Zhuge et al. (2024); LangChain (2024); Hu et al. (2024); Zhang et al. (2024), a task challenging for LLMs when generating workflows. To overcome these challenges, AutoAgent adopts an event-driven approach where we model each agent's task-solving as an event. By leveraging event listening and triggering mechanisms, AutoAgent enables seamless collaboration between agents, offering greater flexibility and adaptability compared to rigid graph structures.

**Constructing New Workflows**. The process of creating a new workflow is itself a multi-agent workflow. The Workflow Form Agent analyzes the requirements and existing tools/agents to determine if new agents need to be created, which agents should form the workflow, and what the listening and triggering logic between events should be. It then generates structured XML code.

During the parsing phase, a robust error detection mechanism verifies whether the generated workflow form complies with system constraints (*e.g.*, constraints on the `on_start` event). If the constraints are not satisfied, detailed error messages are sent back to the Workflow Form Agent as feedback for regeneration. If the constraints are satisfied, the workflow form is passed to the Workflow Editor Agent, which creates new agents (if needed), constructs the new workflow, and executes it on the task. Detailed algorithms and system prompts are shown in Appendix Sec A.6.2.

## 4 EVALUATION

### 4.1 EVALUATION FOR A GENERALIST AGENT SYSTEM

**Dataset and Evaluation Protocols**.

The GAIA benchmark Mialon et al. (2024) evaluates General AI Assistants through 466 test and 165 validation questions across 3 difficulty levels, assessing Reasoning, Multi-Modality Handling, Web Browsing, and Tool-Use Proficiency. We evaluated AutoAgent on GAIA's validation set using success rate as the metric, measuring task completion performance on human-like challenges.

**Baseline Methods**. The baselines we selected are divided into two categories: Open-Source: FRIDAY Wu et al. (2024), Magentic-1 Fourney et al. (2024), Multi-Agent Experiment v0.1 (powered by AutoGen)Microsoft (2024), Hugging-Face AgentsHuggingFace (2024), Langfun Agent Google (2024); Closed-Source: TapeAgent, AgentIM, Trase Agent Trase (2024), Omne, Barcelona[1], and the h2oGPTe Agent H2O.ai (2024). These diverse baselines represent the current state-of-the-art in open-source and proprietary multi-agent systems, providing a comprehensive landscape for evaluating the performance and capabilities of our proposed AutoAgent framework.

Table 1: Performance comparison between the baseline models and our AutoAgent on the GAIA benchmark. The results we report are those published on the GAIA.

| Agent Name | Avg. | L1 | L2 | L3 |
|---|---|---|---|---|
| TapeAgent v0.1 | 33.94 | 47.17 | 34.88 | 3.85 |
| FRIDAY | 34.55 | 45.28 | 34.88 | 11.54 |
| Magentic-1 | 36.97 | 54.72 | 33.72 | 11.54 |
| AgentIM | 37.58 | 50.94 | 36.05 | 15.38 |
| Multi-Agent Exp v0.1 | 39.39 | 54.72 | 38.37 | 11.54 |
| AgentIM v1.1 | 40.00 | 50.94 | 40.70 | 15.38 |
| Trase Agent | 40.00 | 47.17 | 40.70 | 23.08 |
| HuggingFace Agents | 44.24 | 58.49 | 43.02 | 19.23 |
| Magentic-1 (o1) | 46.06 | 56.60 | 46.51 | 23.08 |
| omne | 46.06 | 60.38 | 44.19 | 23.08 |
| Trase Agent v0.2 | 47.27 | 58.49 | 46.51 | 26.92 |
| Barcelona v0.1 | 50.30 | 62.26 | 50.00 | 26.92 |
| Langfun Agent v2.0 | 54.55 | 60.38 | 59.30 | 26.92 |
| h2oGPTe Agent v1.6.8 | 63.64 | 67.92 | 67.44 | 42.31 |
| **AutoAgent** | 55.15 | 71.70 | 53.49 | 26.92 |

**Implementation Details**. To address tasks in the GAIA benchmark, we utilize a combination of the System Utilities of the Model and the Tool Editor Agent from the Agentic-SDK. The basic agents first attempt to complete the task while collecting relevant information and reflections. If successful, the result is directly returned. If not, the Tool Editor Agent creates new tools to continue the task. During validation, Claude-Sonnet-3.5 is used by default.

**Evaluation Results and Analysis**. The results in Table 1 reveal the following key observations:

• **Obs.1. Overall Superiority of AutoAgent**: Our method significantly outperforms all open-source agent systems and achieves performance close to the latest agent system, h2oGPTe Agent v1.6.8 (submitted on December 16, 2024), securing a stable position in the top 2 rankings. Notably,

---

[1]TapeAgent, AgentIM, Omne, and Barcelona are anonymous.

our approach demonstrates superior performance on Level 1 tasks compared to all state-of-the-art baselines, becoming the first method to achieve over 70% accuracy rate. This success is attributed to the well-designed System Utilities and the stable interaction of basic agents with the environment, enabling efficient solutions to everyday simple tasks.

• **Obs.2. Effectiveness of Key Components**: Specifically, our framework demonstrates significantly superior performance compared to Magentic-1 Fourney et al. (2024), a recent representative open-source MAS, and FRIDAY, a classic self-improved framework. While Magentic-1 leverages the powerful reasoning capabilities of o1-preview to design complex Orchestrator Agent (also the Coder Agent), our framework emphasizes the stability of interactions between sub-agents and their respective environments, as well as the precision of tool definitions. Under these conditions, the Orchestrator Agent achieves better results with simple prompts and handoff tools.

## 4.2 EVALUATION OF AUTOAGENT ON THE RETRIEVAL-AUGMENTED GENERATION TASK

**Benchmark Dataset and Evaluation Protocols**. To test the basic functionalities of the AutoAgent, we use the RAG task as the testing benchmark. MultiHop-RAG Tang & Yang (2024) is a dataset designed to evaluate RAG capabilities, requiring the RAG methods to gather information from multiple sources and generate responses, which aligns with the file functionality logic of AutoAgent. We evaluate using two metrics: **Accuracy (Acc)** measures response consistency with expected answers (e.g., "ChatGPT" or "OpenAI's ChatGPT" are both correct for "Which AI tool reached 100M daily users in March?"). **Error (Err)** counts confident but incorrect responses (e.g., answering "Bard" to the above query).

**Baseline Methods**. The baselines represent a diverse range of LLM-based RAG techniques. The chunk methods, such as NaiveRAG Mao et al. (2020) and HyDE Gao et al. (2022), utilize the original text segmentation. The graph methods, including MiniRAG Fan et al. (2025) and LightRAG Guo et al. (2024), manage files as sophisticated graphs. In contrast, Langchain's Agentic RAG LangChain (2023) innovatively accesses files through intelligent software agents. These baselines cover a wide array of strategies for leveraging large language models to retrieve and generate robust responses.

**Implementation Details**. We used gpt-4o-mini OpenAI (2023) as the LLM and text-embedding-3-small for embeddings. We followed MultiHopRAG Tang & Yang (2024) for text chunking, with 256-token chunks and top-6 retrieval. This leverages the gpt-4o-mini's language abilities while text-embedding-3-small provides retrieval, with MultiHopRAG's chunking managing information effectively.

**Evaluation Results and Analysis**. We summarize the key observations from Table 2.

• **Superior Performance of AutoAgent**. The results clearly demonstrate the superior performance of our proposed AutoAgent model compared to other baselines on the Multihop-RAG task. By leveraging a more flexible and adaptive agent-based framework, AutoAgent is able to dynamically orchestrate the retrieval and reasoning process, outperforming even other baselines.

Table 2: Evaluation of AutoAgent for RAG.

| Method | acc | err | acc | err |
|---|---|---|---|---|
| Chunk-Based | NaiveRAG | | HyDE | |
| | 53.36% | 12.28% | 56.59% | 16.55% |
| Graph-Based | MiniRAG | | LightRAG | |
| | 57.81% | 34.78% | 58.18% | 35.40% |
| Agent-Based | Langchain | | **AutoAgent** | |
| | 62.83% | 20.50% | **73.51%** | 14.20% |

• **AutoAgent vs. LangChain**. Our method significantly outperforms LangChain, which is also an agentic RAG. This is due to AutoAgent's more flexible framework, where agents do not need to rely on predefined workflows and tools to execute file search tasks. The proposed model can orchestrate workflows on the fly during the search process, leading to more efficient and accurate results.

## 4.3 AUTOAGENT'S PERFORMANCE ON OPEN-ENDED TASKS

This section thoroughly explores the capabilities of the AutoAgent framework in generating agents and workflows based on even vague, natural language inputs across various scenarios. To illustrate the breadth of AutoAgent's abilities, we will examine its performance on tasks of varying difficulty - from the creation of a single agent to the orchestration of multiple, coordinated agents. (We put the results of single agent in Appendix A.7.1)

**Task with Multi-Agents**. To further validate AutoAgent's capability to generate agents and integrate third-party tools, we tasked it with creating a Financial Agent based on the following requirements:

```
I want to create `Financial Agent` that can help me to do two kinds of
tasks:
```

Table 3: Comparison between single LLMs and the AI-generated Majority Voting workflow.

| Models | gpt-4o 0806 | claude-3.5-sonnet 1022 | deepseek-v3 | Majority Voting Workflow (3 models) |
|--------|-------------|------------------------|-------------|-------------------------------------|
| pass@1 | 66.4 | 66.4 | 74.2 | **75.6** |

```
1. Manage the private financial docs. I have a folder that contain the
 financial docs in my local machine, and I want to help me to manage them.
2. Search the financial information online. You may help me to:
- get balance sheets for a given ticker over a given period.
- get cash flow statements for a given ticker over a given period.
- get income statements for a given ticker over a given period.
```

**Building a Comprehensive Financial Agent**. The Agent Form Agent created two agents: the **Document Manager Agent** and **Market Research Agent**, with XML structure shown in List 18. The Tool Editor Agent created tools: `get_balance_sheet`, `get_cash_flow`, `get_income_statement`, and `analyze_financial_data`. The Agent Editor Agent then composed these agents and established a Financial Analysis Orchestrator for coordination.

The Financial Analysis Orchestrator leveraged new tools and existing capabilities to conduct research on local documents and external data sources. This process produced a comprehensive research report, shown in List 19 in the Appendix. The agent trajectory is detailed in Tab 6 in the Appendix. Despite encountering a SyntaxError during initial creation, the Agent Editor successfully self-debugged and completed the task, demonstrating the robustness of AutoAgent system.

**Workflow Generation**. Scaling Test-Time Compute has been validated as a superior approach for solving reasoning problems. However, manually constructing workflows poses a high barrier to entry. We aim to explore whether AutoAgent's automatic creation of agents and workflows can bridge the gap between the idea of Test-Time Compute and the implementation of workflows. Taking the majority voting method with multiple models as an example:

```
I want to create a workflow that can help me to solving the math problem.
The workflow should:
1. Parallelize solving the math problem with the same `Math Solver
 Agent` using different language models (`gpt-4o`, `claude-3-5-sonnet`,
 `deepseek-chat`)
2. Aggregate the results from the `Math Solver Agent` and return the
 final result using majority voting.
```

**Potential Test-Time Scaling Law**. Upon receiving the requirements, the Workflow Form Agent generated an XML-formatted workflow table (List 20). This table includes two new agents: **Math Solver Agent** and **Vote Aggregator Agent**. After validation, the Agent Editor Agent created agents. The Workflow Editor Agent then constructed a new workflow based on the form and conducted tests. To validate the workflow's practicality, we performed comparative experiments on the MATH-500 dataset Lightman et al. (2024) using 3 LLMs (`gpt-4o-20240806`, `claude-3.5-sonnet-20241022`, `deepseek-v3`) and a Majority Voting workflow. As shown in Tab 3, the generated workflow performs significantly better than state-of-the-art baselines. We selected cases from `deepseek-v3` (Tab 5) where AutoAgent's workflow effectively corrected errors through multi-model collaboration, demonstrating its potential to establish scaling laws in LLMs.

## 5 CONCLUSION

The AutoAgent framework marks a significant advancement in democratizing LLM-powered agent technology, making it accessible to the non-programming majority. By bridging high-level natural language requirements with the practical implementation of multi-agent systems and workflows, MetaChain empowers users to create, customize, and deploy agents, tools, and workflows without requiring substantial technical expertise. Its modular architecture, versatile Agentic System Utilities, and LLM-powered Actionable Engine work together to enable seamless automation of agent development and task execution. Unique features such as the Self-Organizing File System and Self-Play Agent Customization further enhance AutoAgent's capabilities, allowing for dynamic agent evolution and task-specific optimization. Extensive evaluations demonstrate AutoAgent's superior performance, highlighting its transformative potential in making LLM capabilities accessible to a broad user base.

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

# A APPENDIX

In the supplementary materials, we provide a detailed technical description of the 'Agentic System Utilities' implementation within our AutoAgent framework.

## A.1 SYSTEM-LEVEL TOOLS

To empower our diverse array of system-level agents, we have carefully curated and predefined seven distinct categories of powerful tools. These tools span a wide range of functionalities, including coding, web browsing, file management, creating new tools, agents, and workflows, as well as natural language question answering for documents. The detailed names and comprehensive descriptions of these versatile tools are presented in Table 4.

Table 4: List of detailed information of system-level tools.

| Tool Name | Category | Description |
|---|---|---|
| run_python | Coding | Run a python script. |
| execute_command | Coding | Execute a command in the system shell. Use this function when there is a need to run a system command, and execute programs. |
| gen_code_tree_structure | Coding | Generate a tree structure of the code in the specified directory. Use this function when you need to know the overview of the codebase and want to generate a tree structure of the codebase. |
| create_directory | Coding | Create a directory if it does not exist. Use this function when there is a need to create a new directory. |
| list_files | Coding | List all files and directories under the given path if it is a directory. Use this function when there is a need to list the contents of a directory. |
| write_file | Coding | Write content to a file. Use this function when there is a need to write content to an existing file. |
| create_file | Coding | Create a file with the given path and content. Use this function when there is a need to create a new file with initial content. |
| read_file | Coding | Read the contents of a file and return it as a string. Use this function when there is a need to check an existing file. |
| terminal_page_to | Coding | Move the viewport to the specified page index. The index starts from 1. Use this function when you want to move the viewport to a specific page, especially when the middle of terminal output are meaningless, like the output of progress bar or output of generating directory structure when there are many datasets in the directory, you can use this function to move the viewport to the end of terminal where meaningful content is. |

| terminal_page_down | Coding | Scroll the viewport DOWN one page-length in the current terminal. Use this function when the terminal is too long and you want to scroll down to see the next content. |
|---|---|---|
| terminal_page_up | Coding | Scroll the viewport UP one page-length in the current terminal. Use this function when the terminal is too long and you want to scroll up to see the previous content. |
| input_text | Web | Types the given text value into the specified field. |
| click | Web | Clicks the mouse on the target with the given element bid. |
| page_down | Web | Scrolls the entire browser viewport one page DOWN towards the end. |
| page_up | Web | Scrolls the entire browser viewport one page UP towards the beginning. |
| history_back | Web | Navigates back one page in the browser's history. This is equivalent to clicking the browser back button. |
| history_forward | Web | Navigates forward one page in the browser's history. This is equivalent to clicking the browser forward button. |
| visit_url | Web | Navigate directly to a provided URL using the browser's address bar. Prefer this tool over other navigation techniques in cases where the user provides a fully-qualified URL (e.g., choose it over clicking links, or inputing queries into search boxes). |
| web_search | Web | Performs a web search on 'https://www.google.com.sg/?hl=en&gl=US' with the given query. |
| sleep | Web | Wait a short period of time. Call this function if the page has not yet fully loaded, or if it is determined that a small delay would increase the task's chances of success. |
| get_page_markdown | Web | Get the markdown content of the current page. Use this tool if you need to watch the Youtube video, Wikipedia page, or other pages that contain media content. Note that this tool can only be used after you have visited a valid page. |
| open_local_file | File | Open a local file at a path in the text-based browser and return current viewport content. |
| page_up_markdown | File | Scroll the viewport UP one page-length in the current file and return the new viewport content. |
| page_down_markdown | File | Scroll the viewport DOWN one page-length in the current file and return the new viewport content. |
| find_next | File | Scroll the viewport to next occurrence of the search string. |

| visual_question_answering | File | This tool is used to answer questions about attached images or videos. |
|---|---|---|
| find_on_page_ctrl_f | File | Scroll the viewport to the first occurrence of the search string. This is equivalent to Ctrl+F. |
| list_tools | Tools Edit | List all plugin tools in the MetaChain. |
| create_tool | Tools Edit | Create a plugin tool. |
| delete_tool | Tools Edit | Delete a plugin tool. |
| run_tool | Tools Edit | Run a tool with the given code. |
| search_trending_models_on_huggingface | Tools Edit | Search trending models on Hugging Face. Use this tool when you want to create a tool that uses Hugging Face models, only support the following tags: ['audio-text-to-text', 'text-to-image', 'image-to-image', 'image-to-video', 'text-to-video', 'text-to-speech', 'text-to-audio', 'automatic-speech-recognition', 'audio-to-audio']. |
| get_hf_model_tools_doc | Tools Edit | Get the detailed information of a model on Hugging Face, such as the detailed usage of the model containing the model's README.md. You should use this tool after you have used 'search_trending_models_on_huggingface' to find the model you want to use. |
| get_api_plugin_tools_doc | Tools Edit | Retrieve satisfied tool documents based on the query text. |
| list_agents | Agents Edit | List all plugin agents in the MetaChain. |
| read_agents | Agents Edit | Get detailed information of plugin agents in the MetaChain. |
| delete_agent | Agents Edit | Delete a plugin agent. |
| run_agent | Agents Edit | Run a plugin agent. |
| create_agent | Agents Edit | Use this tool to create a new agent or modify an existing agent. |
| create_orchestrator_agent | Agents Edit | Use this tool to create a orchestrator agent for the given sub-agents. You MUST use this tool when you need to create TWO or MORE agents and regard them as a whole to complete a task. |
| create_workflow | Workflows Edit | Create a workflow. |
| list_workflows | Workflows Edit | List all workflows in the MetaChain. |
| run_workflow | Workflows Edit | Run a workflow. |

| save_raw_docs_to_vector_db | RAG | Save the raw documents to the vector database. The documents could be: - ANY text document with the extension of pdf, docx, txt, etcs. - A zip file containing multiple text documents - a directory containing multiple text documents All documents will be converted to raw text format and saved to the vector database in the chunks of 4096 tokens. |
|---|---|---|
| query_db | RAG | Retrieve information from the database. Use this function when you need to search for information in the database. |
| modify_query | RAG | Modify the query based on what you know. Use this function when you need to modify the query to search for more relevant information. |
| answer_query | RAG | Answer the user query based on the supporting documents. |
| can_answer | RAG | Check if you have enough information to answer the user query. |

## A.2 WEB AGENT

The specific tools and system prompt for implementing the Web Agent are as follows:

Listing 1: Tools of **Web Agent**

```
[click, page_down, page_up, history_back, history_forward,
web_search, input_text, sleep, visit_url, get_page_markdown,
transfer_back_to_orchestrate_agent]
```

Listing 2: System Prompt of **Web Agent**

```
Review the current state of the page and all other information to
↪ find the best possible next action to accomplish your goal. Your
↪ answer will be interpreted and executed by a program, make sure to
↪ follow the formatting instructions.
Note that if you want to analyze the YouTube video, Wikipedia page,
↪ or other pages that contain media content, or you just want to
↪ analyze the text content of the page in a more detailed way, you
↪ should use `get_page_markdown` tool to convert the page information
↪ to markdown text. And when browsing the web, if you have downloaded
↪ any files, the path of the downloaded files will be
↪ `/workplace/downloads`, and you CANNOT open the downloaded files
↪ directly, you should transfer back to the `System Orchestrate
↪ Agent`, and let `System Orchestrate Agent` to transfer to `Local
↪ File Agent` to open the downloaded files.
When you think you have completed the task the `System Orchestrate
↪ Agent` asked you to do, you should use
↪ `transfer_back_to_orchestrate_agent` to transfer the conversation
↪ back to the `System Orchestrate Agent`. And you should not stop to
↪ try to solve the user's request by transferring to `System
↪ Orchestrate Agent` only until the task is completed.
```

## A.3 LOCAL FILE AGENT

The Local File Agent is equipped with a tailored set of tools and system prompts to enable it to efficiently manage and interact with files and directories. This specialized toolkit includes:

Listing 3: Tools of **Local File Agent**

```
[open_local_file, page_up_markdown, page_down_markdown,
find_on_page_ctrl_f, find_next, visual_question_answering,
transfer_back_to_orchestrate_agent]
```

Listing 4: System Prompt of **Local File Agent**

```
You are a file surfer agent that can handle local files.

You can only access the files in the folder `/workplace` and when you
↪ want to open a file, you should use absolute path from root like
↪ `/workplace/...`.

Note that `open_local_file` can read a file as markdown text and ask
↪ questions about it. And `open_local_file` can handle the following
↪ file extensions: [".html", ".htm", ".xlsx", ".pptx", ".wav",
↪ ".mp3", ".flac", ".pdf", ".docx"], and all other types of text
↪ files.

But IT DOES NOT HANDLE IMAGES, you should use
↪ `visual_question_answering` to see the image.

If the converted markdown text has more than 1 page, you can use
↪ `page_up`, `page_down`, `find_on_page_ctrl_f`, `find_next` to
↪ navigate through the pages.

When you think you have completed the task the `System Orchestrate
↪ Agent` asked you to do, you should use
↪ `transfer_back_to_orchestrate_agent` to transfer the conversation
↪ back to the `System Orchestrate Agent`. And you should not stop to
↪ try to solve the user's request by transferring to `System
↪ Orchestrate Agent` only until the task is completed.

If you are unable to open the file, you can transfer the conversation
↪ back to the `System Orchestrate Agent`, and let the `Coding Agent`
↪ try to solve the problem by coding.
```

## A.4 CODING AGENT

The specific tools and system prompts for implementing the Coding Agent are as follows:

Listing 5: Tools of **Coding Agent**

```
[gen_code_tree_structure, execute_command, read_file, create_file,
write_file, list_files, create_directory, run_python,
terminal_page_up, terminal_page_down, terminal_page_to,
transfer_back_to_orchestrate_agent]
```

Listing 6: System Prompt of **Coding Agent**

```
You are a helpful programming assistant that can write and execute
↪ code. You are working in the folder: `/workplace`, and you can only
↪ access the files in this folder.
Your can leverage your capabilities by using the specific functions
↪ listed below:
1. Creating project structures based on the user requirement using
↪ function `create_directory`.
2. Writing clean, efficient, and well-documented code using function
↪ `create_file` and `write_file`.
3. You must run python scripts using function `run_python` rather
↪ than using the `execute_command` function.
4. Exam the project to re-use the existing code snippets as much as
↪ possible, you may need to use
```

```
functions like `list_files`, `read_file` and `write_file`.
5. Writing the code into the file when creating new files, do not
↪ create empty files.
6. Before you write code into the existing files, you should first
↪ read the file content using function `read_file` and reserve the
↪ original content as much as possible.
7. Decide whether the task requires execution and debugging before
↪ moving to the next or not.
8. Generate the commands to run and test the current task, and the
↪ dependencies list for this task.
9. You only write Python scripts, don't write Jupiter notebooks which
↪ require interactive execution.
10. Note that every path you read, write, or search should be the
↪ absolute path (starting with "/").
11. If you should use programming other than Python, you should use
↪ the `write_file` function to write the code into a file, and then
↪ use the `execute_command` function to run the code.
12. If the terminal output is too long, you should use
↪ `terminal_page_up` to move the viewport up, `terminal_page_down` to
↪ move the viewport down, `terminal_page_to` to move the viewport to
↪ the specific page of terminal where the meaningful content is.

Note that you can use this agent to make complex computation, write a
↪ api request, and anything else that can be done by writing code.

When you think you have completed the task the `System Orchestrate
↪ Agent` asked you to do, you should use
↪ `transfer_back_to_orchestrate_agent` to transfer the conversation
↪ back to the `System Orchestrate Agent`. And you should not stop to
↪ try to solve the user's request by transferring to `System
↪ Orchestrate Agent` only until the task is completed.

[IMPORTANT] You can only complete the task by coding. Talk is cheap,
↪ show me the code with tools.
```

## A.5 ORCHESTRATOR AGENT

The specific tools and system prompt for implementing the Orchestrator Agent are as follows:

Listing 7: Tools of **Orchestrator Agent**

```
[transfer_to_local_file_agent, transfer_to_web_agent,
transfer_to_coding_agent]
```

Listing 8: System Prompt of **Orchestrator Agent**

```
You are a helpful assistant that can help the user with their request.
Based on the state of solving user's task, your responsibility is to
↪ determine which agent is best suited to handle the user's request
↪ under the current context, and transfer the conversation to that
↪ agent. And you should not stop to try to solve the user's request
↪ by transferring to another agent only until the task is completed.

There are three agents you can transfer to:
1. use `transfer_to_local_file_agent` to transfer to `Local File
↪ Agent`, it can help you to open any type of local files and browse
↪ the content of them.
2. use `transfer_to_web_agent` to transfer to `Web Agent`, it can
↪ help you to open any website and browse any content on it.
3. use `transfer_to_coding_agent` to transfer to `Coding Agent`, it
↪ can help you to write code to solve the user's request, especially
↪ some complex tasks.
```

## A.6 DETAILED IMPLEMENTATION OF "SELF-PLAY AGENT CUSTOMIZATION" IN AUTOAGENT

### A.6.1 AGENT CREATION WITHOUT WORKFLOW

The following details demonstrate the specific process of Agent Creation without Workflow (Alg 1), as well as the tools and system prompts used in the implementation of Agent Profiling Agent, Tool Editor Agent, and Agent Editor Agent.

---

**Algorithm 1** Controllable Workflow of Creating Tools and Agents

---

1: **Input:** requirements $\mathcal{R}$, existing tool set $\mathcal{A}$, existing agent set $\pi$, task $\mathcal{T}$ (optional), Maximum iterations of attempts $\mathbf{M}$.
2: **Output:** the response of requirements for creating tools and agents to solve the task.
3: $\texttt{AgentProfile} = \textbf{agent\_profile\_agent}(\mathcal{R}|\mathcal{A}, \pi)$
4: $\texttt{ParsingResults}_0 = \textbf{form\_parsing\_function}(\texttt{AgentProfile}|\mathcal{R}, \mathcal{A}, \pi)$
5: **for** $i = 1$ **to** $\mathbf{M} - 1$ **do**
6:    **if** $\texttt{ParsingResults}_{i-1}$ is 'Success' **then**
7:       **break**
8:    **else**
9:       $\texttt{AgentProfile} = \textbf{agent\_profile\_agent}(\mathcal{R}, \texttt{ParsingResults}_{i-1}|\mathcal{A}, \pi)$
10:       $\texttt{ParsingResults}_i = \textbf{form\_parsing\_function}(\texttt{AgentProfile}|\mathcal{R}, \mathcal{A}, \pi)$
11:    **end if**
12: **end for**
13: **if** There are new tools need to be created in $\texttt{AgentProfile}$ **then**
14:    // Automatically create tools and test them.
15:    $\texttt{ToolsResults}_0 = \textbf{tool\_editor\_agent}(\texttt{AgentProfile}|\mathcal{R}, \mathcal{A}, \pi)$
16:    **for** $i = 1$ **to** $\mathbf{M} - 1$ **do**
17:       **if** $\texttt{ToolsResults}_{i-1}$ is 'Success' **then**
18:          **break**
19:       **else**
20:          $\texttt{ToolsResults}_i = \textbf{tool\_editor\_agent}(\texttt{AgentProfile}, \texttt{ToolsResults}_{i-1}|\mathcal{R}, \mathcal{A}, \pi)$
21:       **end if**
22:    **end for**
23: **end if**
24: // Automatically create agents and run them on the given task.
25: $\texttt{AgentsResults}_0 = \textbf{agent\_editor\_agent}(\texttt{AgentProfile}, \mathcal{T}|\mathcal{R}, \mathcal{A}, \pi)$
26: **for** $i = 1$ **to** $\mathbf{M} - 1$ **do**
27:    **if** $\texttt{AgentsResults}_{i-1}$ is 'Success' **then**
28:       **break**
29:    **else**
30:       $\texttt{AgentsResults}_i = \textbf{agent\_editor\_agent}(\texttt{AgentProfile}, \mathcal{T}, \texttt{AgentsResults}_{i-1}|\mathcal{R}, \mathcal{A}, \pi)$
31:    **end if**
32: **end for**

---

Listing 9: System Prompt of **Agent Profiling Agent**

```
You are an agent specialized in creating agent forms for the
↪ MetaChain framework.

Your task is to analyze user requests and generate structured
↪ creation forms for either single or multi-agent systems.

KEY COMPONENTS OF THE FORM:
1. <agents> - Root element containing all agent definitions

2. <system_input> - Defines what the system receives
   - Must describe the overall input that the system accepts
   - For single agent: Same as agent_input
   - For multi-agent: Should encompass all possible inputs that will
   ↪ be routed to different agents

3. <system_output> - Specifies system response format
```

```
      - Must contain exactly ONE key-description pair
     - <key>: Single identifier for the system's output
     - <description>: Explanation of the output
     - For single agent: Same as agent_output
     - For multi-agent: Should represent the unified output format from
     ↪ all agents

4. <agent> - Individual agent definition
   - name: Agent's identifier
   - description: Agent's purpose and capabilities
   - instructions: Agent's behavioral guidelines
     * To reference global variables, use format syntax: {variable_key}
     * Example: "Help_the_user_{user_name}_with_his/her_request"
     * All referenced keys must exist in global_variables
   - tools: Available tools (existing/new)
   - agent_input:
     * Must contain exactly ONE key-description pair
     * <key>: Identifier for the input this agent accepts
     * <description>: Detailed explanation of the input format
   - agent_output:
     * Must contain exactly ONE key-description pair
     * <key>: Identifier for what this agent produces
     * <description>: Detailed explanation of the output format

5. <global_variables> - Shared variables across agents (optional)
   - Used for constants or shared values accessible by all agents
   - Variables defined here can be referenced in instructions using
   ↪ {key}
   - Example:
     ```xml
     <global_variables>
         <variable>
             <key>user_name</key>
             <description>The name of the user</description>
             <value>John Doe</value>
         </variable>
     </global_variables>
     ```
   - Usage in instructions: "You_are_a_personal_assistant_for_
   ↪ {user_name}."

IMPORTANT RULES:
- For single agent systems:
  * system_input/output must match agent_input/output exactly
- For multi-agent systems:
  * system_input should describe the complete input space
  * Each agent_input should specify which subset of the system_input
  ↪ it handles
  * system_output should represent the unified response format

Existing tools you can use is:
...

Existing agents you can use is:
...

EXAMPLE 1 - SINGLE AGENT:

User: I want to build an agent that can answer the user's question
↪ about the OpenAI products. The document of the OpenAI products is
↪ available at '/workspace/docs/openai_products/'.
The agent should be able to:
1. query and answer the user's question about the OpenAI products
↪ based on the document.
```

```
1080  2. send email to the user if the sending email is required in the
1081  ↪ user's request.
1082
1083  The form should be:
1084  <agents>
1085     <system_input>
1086        Questions from the user about the OpenAI products. The document
1086        ↪ of the OpenAI products is available at
1087        ↪ '/workspace/docs/openai_products/'.
1088     </system_input>
1089     <system_output>
1090        <key>answer</key>
1090        <description>The answer to the user's question.</description>
1091     </system_output>
1092     <agent>
1093        <name>Helper Center Agent</name>
1094        <description>The helper center agent is an agent that serves as
1095        ↪ a helper center agent for a specific user to answer the
1095        ↪ user's question about the OpenAI products.</description>
1096        <instructions>You are a helper center agent that can be used to
1097        ↪ help the user with their request.</instructions>
1098        <tools category="existing">
1099           <tool>
1100              <name>save_raw_docs_to_vector_db</name>
1101              <description>Save the raw documents to the vector
1101              ↪ database. The documents could be:
1102              - ANY text document with the extension of pdf, docx, txt,
1103              ↪ etcs.
1104              - A zip file containing multiple text documents
1105              - a directory containing multiple text documents
1105              All documents will be converted to raw text format and
1106              ↪ saved to the vector database in the chunks of 4096
1107              ↪ tokens.</description>
1108           </tool>
1109           <tool>
1110              <name>query_db</name>
1111              <description>Query the vector database to find the answer
1111              ↪ to the user's question.</description>
1112           </tool>
1113           <tool>
1114              <name>modify_query</name>
1115              <description>Modify the user's question to a more
1115              ↪ specific question.</description>
1116           </tool>
1117           <tool>
1118              <name>answer_query</name>
1119              <description>Answer the user's question based on the
1119              ↪ answer from the vector database.</description>
1120           </tool>
1121           <tool>
1122              <name>can_answer</name>
1123              <description>Check if the user's question can be answered
1124              ↪ by the vector database.</description>
1124           </tool>
1125        </tools>
1126        <tools category="new">
1127           <tool>
1128              <name>send_email</name>
1129              <description>Send an email to the user.</description>
1129           </tool>
1130        </tools>
1131        <agent_input>
1132           <key>user_question</key>
1133           <description>The question from the user about the OpenAI
1133           ↪ products.</description>
```

```
1134          </agent_input>
1135          <agent_output>
1136              <key>answer</key>
1137              <description>The answer to the user's question.</description>
1138          </agent_output>
1139      </agent>
1140  </agents>

1141  EXAMPLE 2 - MULTI-AGENT:

1142
1143  User: I want to build a multi-agent system that can handle two types
1144  ↪ of requests for the specific user:
1145  1. Purchase a product or service
1146  2. Refund a product or service
1147  The specific user worked for is named John Doe.

1148  The form should be:
1149  <agents>
1150      <system_input>
1151          The user request from the specific user about the product or
1152          ↪ service, mainly categorized into 2 types:
1153          - Purchase a product or service
1154          - Refund a product or service
1155      </system_input>
1156      <system_output>
1157          <key>response</key>
1158          <description>The response of the agent to the user's
1159          ↪ request.</description>
1160      </system_output>
1161      <global_variables>
1162          <variable>
1163              <key>user_name</key>
1164              <description>The name of the user.</description>
1165              <value>John Doe</value>
1166          </variable>
1167      </global_variables>
1168      <agent>
1169          <name>Personal Sales Agent</name>
1170          <description>The personal sales agent is an agent that serves
1171          ↪ as a personal sales agent for a specific user.</description>
1172          <instructions>You are a personal sales agent that can be used
1173          ↪ to help the user {user_name} with their
1174          ↪ request.</instructions>
1175          <tools category="new">
1176              <tool>
1177                  <name>recommend_product</name>
1178                  <description>Recommend a product to the
1179                  ↪ user.</description>
1180              </tool>
1181              <tool>
1182                  <name>recommend_service</name>
1183                  <description>Recommend a service to the
1184                  ↪ user.</description>
1185              </tool>
1186              <tool>
1187                  <name>conduct_sales</name>
1188                  <description>Conduct sales with the user.</description>
1189              </tool>
1190          </tools>
1191          <agent_input>
1192              <key>user_request</key>
1193              <description>Request from the specific user for purchasing a
1194              ↪ product or service.</description>
1195          </agent_input>
1196          <agent_output>
```

```
1188            <key>response</key>
1189            <description>The response of the agent to the user's
1190            ↪ request.</description>
1191         </agent_output>
1192      </agent>
1193      <agent>
1194         <name>Personal Refunds Agent</name>
1195         <description>The personal refunds agent is an agent that serves
              ↪ as a personal refunds agent for a specific user.</description>
1196         <instructions>Help the user {user_name} with a refund. If the
1197         ↪ reason is that it was too expensive, offer the user a
1198         ↪ discount. If they insist, then process the
              ↪ refund.</instructions>
1199         <tools category="new">
1200            <tool>
1201               <name>process_refund</name>
1202               <description>Refund an item. Refund an item. Make sure
1203               ↪ you have the item_id of the form item_... Ask for user
                  ↪ confirmation before processing the refund.</description>
1204            </tool>
1205            <tool>
1206               <name>apply_discount</name>
1207               <description>Apply a discount to the user's
1208               ↪ cart.</description>
1209            </tool>
              </tools>
1210         <agent_input>
1211            <key>user_request</key>
1212            <description>Request from the specific user for refunding a
1213            ↪ product or service.</description>
1214         </agent_input>
1215         <agent_output>
1216            <key>response</key>
1217            <description>The response of the agent to the user's
              ↪ request.</description>
1218         </agent_output>
1219      </agent>
       </agents>
1220
1221   GUIDELINES:
1222   1. Each agent must have clear, focused responsibilities
1223   2. Tool selections should be minimal but sufficient
1224   3. Instructions should be specific and actionable
1225   4. Input/Output definitions must be precise
       5. Use global_variables for shared context across agents
1226
1227   Follow these examples and guidelines to create appropriate agent
1228   ↪ forms based on user requirements.
1229
1230
1231
```

Listing 10: Tools of **Tool Editor Agent**

```
[list_tools, create_tool, run_tool, delete_tool,
get_api_plugin_tools_doc, execute_command, terminal_page_down,
terminal_page_up, terminal_page_to,
search_trending_models_on_huggingface, get_hf_model_tools_doc]
```

Listing 11: System Prompt of **Tool Editor Agent**

```
You are a tool editor agent responsible for managing plugin tools in
↪ the MetaChain framework. Your core responsibility is to edit,
↪ create, and manage plugin tools that can be used by other agents.

[PLUGIN TOOLS SYSTEM]
```

```
1242   - Plugin tools are the building blocks of MetaChain
1243   - All available plugin tools are as follows:
1244   ...
1245   - Plugin tools can ONLY be executed using `run_tool(tool_name,
1246 ↪ run_code)`. You should import `run_tool` by `from metachain.tools
1247 ↪ import run_tool`.
       - NEVER try to import and run plugin tools directly - always use
1248 ↪ `run_tool`
1249
1250   [TOOL CREATION WORKFLOW]
1251   1. ALWAYS start with `list_tools()` to check existing tools
1252
1253   2. For NEW plugin tool creation, FOLLOW THIS ORDER:
         a. For third-party API integration (e.g., RapidAPI, external
1254 ↪ services):
1255       - MUST FIRST use `get_api_plugin_tools_doc` to get API
1256 ↪ documentation and keys
1257       - API keys should be embedded IN the function body, NOT as
      ↪ parameters.
1258       - The API keys are always in the retrieved information from
1259 ↪ `get_api_plugin_tools_doc`, DO NOT guess the API keys by
1260 ↪ yourself.
1261       - Follow the API implementation details from the documentation
1262
1263       b. For modal transformation tasks (image/video/audio
      ↪ generation/processing):
1264       - FIRST use `search_trending_models_on_huggingface` to find
1265 ↪ suitable models, only support the following tags:
1266 ↪ ['audio-text-to-text', 'text-to-image', 'image-to-image',
      ↪ 'image-to-video', 'text-to-video', 'text-to-speech',
1267 ↪ 'text-to-audio', 'automatic-speech-recognition',
1268 ↪ 'audio-to-audio'].
1269       - Then use `get_hf_model_tools_doc` for detailed model information
1270       - Only use internal knowledge if no suitable models are found
1271
1272       c. For visual analysis tasks (images/videos):
1273       - MUST use the existing `visual_question_answering` plugin tool
      ↪ by `run_tool("visual_question_answering", "from
1274 ↪ metachain.tools import visual_question_answering; ...")`. DO
1275 ↪ NOT use it directly without `run_tool`.
1276       - NO direct implementation of visual processing
1277       - Chain with other tools as needed
1278
       3. Plugin Tool Implementation Requirements:
1279       - Use @register_plugin_tool decorator (REQUIRED). You should import
1280 ↪ `register_plugin_tool` by `from metachain.registry import
1281 ↪ register_plugin_tool`.
         - Follow this template:
1282   ```python
1283   ...
1284   ```
1285       - Include clear type hints
1286       - Make tools abstract and reusable
1287       - Use generic names (e.g., 'process_media' not
      ↪ 'process_youtube_video')
1288       - Handle dependencies with `execute_command`
1289
1290   [AVAILABLE TOOLS]
1291   1. get_api_plugin_tools_doc:
1292       - PRIMARY tool for third-party API integration
         - MUST be used FIRST for Finance, Entertainment, eCommerce, etc.
1293       - Provides API documentation AND authentication keys
1294       - API keys should be embedded in tool implementation
1295
       2. search_trending_models_on_huggingface:
```

```
   - Use for finding models for media transformation tasks
   - Supported tags: ['text-to-image', 'image-to-image',
   ↪ 'text-to-video', etc.]
   - Use AFTER checking no suitable API exists via
   ↪ `get_api_plugin_tools_doc`

3. get_hf_model_tools_doc:
   - Get the detailed information of a model on Hugging Face, such as
   ↪ the detailed usage of the model containing the model's README.md.
   - You should use this tool after you have used
   ↪ `search_trending_models_on_huggingface` to find the model you
   ↪ want to use.

4. Other management tools:
   - list_tools(): Check existing tools
   - create_tool(tool_name, tool_code): Create new tools
   - run_tool(tool_name, run_code): REQUIRED method to execute any
   ↪ plugin tool
   - delete_tool(tool_name): Remove tools
   - execute_command: Install dependencies. Handles system-level
   ↪ operations
   - terminal_page_* tools: Navigate long outputs

5. case_resolved & case_not_resolved:
   - case_resolved: after you have created all the tools and tested
   ↪ them using `run_tool` successfully (with the expected output
   ↪ rather than just run it), you should use the `case_resolved`
   ↪ tool to brief the result.
   - case_not_resolved: after you have tried your best to create the
   ↪ tools but failed, you should use the `case_not_resolved` tool to
   ↪ tell the failure reason.

[CRITICAL RULES]
1. Tool Creation Priority:
   - FIRST: Check existing tools via list_tools()
   - SECOND: Use `get_api_plugin_tools_doc` for API-based tools
   - THIRD: Use `search_trending_models_on_huggingface` for media tasks
   - LAST: Use internal knowledge if no other options available

2. API Implementation:
   - NEVER expose API keys as parameters
   - ALWAYS embed API keys in function body
   - Get keys from `get_api_plugin_tools_doc`

3. Tool Design:
   - Tools MUST be abstract, modular, and reusable:
     - Use generic function names (e.g., `download_media` instead of
     ↪ `download_youtube_video`)
     - Break complex tasks into smaller, reusable components
     - Avoid task-specific implementations
     - Use parameters instead of hardcoded values
   - Include proper error handling

[TESTING]
Test new tools using `run_tool`:
`run_tool(tool_name="your_tool", run_code="from metachain.tools
↪ import your_tool; print(your_tool(param1='value1'))")`
```

Listing 12: Tools of **Agent Editor Agent**

```
[list_agents, create_agent, delete_agent, run_agent, execute_command,
read_agent, create_orchestrator_agent, terminal_page_down,
terminal_page_up, terminal_page_to]
```

Listing 13: System Prompt of **Agent Editor Agent**

```
You are an Agent Creator specialized in the MetaChain framework. Your
↪ primary responsibility is to create, manage, and orchestrate agents
↪ based on XML-formatted agent forms.

CORE RESPONSIBILITIES:
1. Parse and implement agent forms
2. Create and manage individual agents
3. Orchestrate multi-agent systems
4. Handle dependencies and system requirements

AVAILABLE FUNCTIONS:
1. Agent Management:
  - `create_agent`: Create new agents or update existing ones
  ↪ strictly following the given agent form.
  - `read_agent`: Retrieve existing agent definitions. Note that if
  ↪ you want to use `create_agent` to update an existing agent, you
  ↪ MUST use the `read_agent` function to get the definition of the
  ↪ agent first.
  - `delete_agent`: Remove unnecessary agents.
  - `list_agents`: Display all available agents and their information.
  - `create_orchestrator_agent`: Create orchestrator for multi-agent
  ↪ systems. If the request is to create MORE THAN ONE agent, after
  ↪ you create ALL required agents, you MUST use the
  ↪ `create_orchestrator_agent` function to create an orchestrator
  ↪ agent that can orchestrate the workflow of the agents. And then
  ↪ use the `run_agent` function to run the orchestrator agent to
  ↪ complete the user task.

2. Execution:
  - run_agent: Execute agent to complete the user task. The agent
  ↪ could be a single agent (single agent form) or an orchestrator
  ↪ agent (multi-agent form).
  - execute_command: Handle system dependencies and requirements
  - terminal_page_down: Move the terminal page down when the terminal
  ↪ output is too long.
  - terminal_page_up: Move the terminal page up when the terminal
  ↪ output is too long.
  - terminal_page_to: Move the terminal page to the specific page
  ↪ when the terminal output is too long, and you want to move to
  ↪ the specific page with the meaningful content.

WORKFLOW GUIDELINES:

1. Single Agent Implementation:
  - Carefully read the agent form and understand the requirements.
  - Create/update agent using create_agent
  - Execute task using run_agent
  - Monitor and handle any errors

2. Multi-Agent Implementation:
  - Create all required agents individually using `create_agent`
  - MUST create an orchestrator agent using
  ↪ `create_orchestrator_agent`
  - Execute task through the `run_agent` function to execute the
  ↪ created orchestrator agent
  - Monitor system performance

3. Error Handling:
  - Check for missing dependencies using `execute_command`
  - Install required packages using execute_command
  - Validate agent creation and execution
  - Report any issues clearly
```

```
BEST PRACTICES:
1. Always verify existing agents using `read_agent` before updates
2. Create orchestrator agents for ANY multi-agent scenario using
↪ `create_orchestrator_agent`
3. Handle dependencies proactively using `execute_command`
4. Maintain clear documentation of created agents
5. Follow the exact specifications from the agent form XML

Remember: Your success is measured by both the accurate creation of
↪ agents and their effective execution of the given tasks.
```

### A.6.2 AGENT CREATION WITH WORKFLOW

The following details demonstrate the specific process of Agent Creation with Workflow (Alg 2), as well as the tools and system prompts used in the implementation of Workflow Profiling Agent and Workflow Editor Agent.

---

**Algorithm 2** Controllable Workflow of Creating Agents and Workflows

---

1: **Input:** requirements $\mathcal{R}$, existing tool set $\mathcal{A}$, existing agent set $\pi$, existing workflow set $\mathcal{W}$ task $\mathcal{T}$ (optional), Maximum iterations of attempts $\mathbf{M}$.
2: **Output:** the response of requirements for creating workflows to solve the task.
3: $\texttt{WorkflowProfile} = \textbf{workflow\_profiling\_agent}(\mathcal{R}|\mathcal{A}, \pi, \mathcal{W})$
4: $\texttt{ParsingResults}_0 = \textbf{form\_parsing\_function}(\texttt{WorkflowProfile}|\mathcal{R}, \mathcal{A}, \pi, \mathcal{W})$
5: **for** $i = 1$ **to** $\mathbf{M} - 1$ **do**
6:    **if** $\texttt{ParsingResults}_{i-1}$ is 'Success' **then**
7:       **break**
8:    **else**
9:       $\texttt{WorkflowProfile} = \textbf{workflow\_profiling\_agent}(\mathcal{R}, \texttt{ParsingResults}_{i-1}|\mathcal{A}, \pi)$
10:       $\texttt{ParsingResults}_i = \textbf{form\_parsing\_function}(\texttt{WorkflowProfile}|\mathcal{R}, \mathcal{A}, \pi, \mathcal{W})$
11:    **end if**
12: **end for**
13: // Automatically create workflows and run them on the given task.
14: $\texttt{WorkflowsResults}_0 = \textbf{workflow\_editor\_agent}(\texttt{WorkflowProfile}, \mathcal{T}|\mathcal{R}, \mathcal{A}, \pi, \mathcal{W})$
15: **for** $i = 1$ **to** $\mathbf{M} - 1$ **do**
16:    **if** $\texttt{WorkflowsResults}_{i-1}$ is 'Success' **then**
17:       **break**
18:    **else**
19:       $\texttt{WorkflowsResults}_i = \textbf{workflow\_editor\_agent}(\texttt{WorkflowProfile}, \mathcal{T}, \texttt{WorkflowsResults}_{i-1}|\mathcal{R}$
20:    **end if**
21: **end for**

---

Listing 14: System Prompt of **Workflow Profiling Agent**

```
You are an agent specialized in creating workflow forms for the
↪ MetaChain framework.

Your task is to analyze user requests and generate structured
↪ creation forms for workflows consisting of multiple agents.

KEY COMPONENTS OF THE FORM:
1. <workflow> - Root element containing the entire workflow definition

2. <name> - The name of the workflow. It should be a single word with
↪ '_' as the separator, and as unique as possible to describe the
↪ speciality of the workflow.

3. <system_input> - Defines what the system receives
  - Must describe the overall input that the system accepts
  - <key>: Single identifier for the input, could be a single word
  ↪ with '_' as the separator.
```

```
   - <description>: Detailed explanation of input format

4. <system_output> - Specifies system response format
   - Must contain exactly ONE key-description pair
   - <key>: Single identifier for the system's output, could be a
   ↪ single word with '_' as the separator.
   - <description>: Explanation of the output format

5. <agents> - Contains all agent definitions
   - Each <agent> can be existing or new (specified by category
   ↪ attribute)
   - name: Agent's identifier
   - description: Agent's purpose and capabilities
   - tools: (optional): Only required for new agents when specific
   ↪ tools are requested
    * Only include when user explicitly requests certain tools

6. <global_variables> - Shared variables across agents in the
↪ workflow (optional)
   - Used for constants or shared values accessible by all agents in
   ↪ EVERY event in the workflow
   - Example:
    ```xml
    <global_variables>
        <variable>
            <key>user_name</key>
            <description>The name of the user</description>
            <value>John Doe</value>
        </variable>
    </global_variables>
    ```

7. <events> - Defines the workflow execution flow
   Each <event> contains:
   - name: Event identifier
   - inputs: What this event receives, should exactly match with the
   ↪ output keys of the events it's listening to
    * Each input has:
     - key: Input identifier (should match an output key from
     ↪ listened events)
     - description: Input explanation
   - task: What this event should accomplish
   - outputs: Possible outcomes of this event
    * Each output has:
     - action: What happens after. Every action has a type and a
     ↪ optional value. Action is categorized into 3 types:
      - RESULT: The event is successful, and the workflow will
      ↪ continue to the next event which is listening to this event.
      ↪ Value is the output of this event.
      - ABORT: The event is not successful, and the workflow will
      ↪ abort. Value could be empty.
      - GOTO: The event is not successful, and the workflow will wait
      ↪ for the next event. Value is the name of the event to go to.
      ↪ The event go to should NOT listen to this event.
     - key: Output identifier (be a single word with '_' as the
     ↪ separator)
     - description: Output explanation
     - condition: when the output occurs, the action will be executed
    * Can have single or multiple outputs:
     - For single output (simple flow):
     ```xml
     <outputs>
        <output>
            <key>result_key</key>
```

```xml
                <description>Description of the result</description>
                <action>
                    <type>RESULT</type>
                </action>
            </output>
        </outputs>
        ```
        - For multiple outputs (conditional flow):
        ```xml
        <outputs>
            <output>
                <key>success_result</key>
                <description>Output when condition A is met</description>
                <condition>When condition A is true</condition>
                <action>
                    <type>RESULT</type>
                </action>
            </output>
            <output>
                <key>should_repeat</key>
                <description>Output when condition B is met</description>
                <condition>When condition B is true</condition>
                <action>
                    <type>GOTO</type>
                    <value>target_event</value>
                </action>
            </output>
            <output>
                <key>failure_result</key>
                <description>Output when condition C is met</description>
                <condition>When condition C is true</condition>
                <action>
                    <type>ABORT</type>
                </action>
            </output>
        </outputs>
        ```
    - listen: Which events trigger this one.
    - agent: Which agent handles this event. Every agent has the name
    ↪ of the agent, and the exact model of the agent (like
    ↪ `claude-3-5-sonnet-20241022` or others)

IMPORTANT RULES:
0. The `on_start` event is a special event that:
    - Must be the first event in the workflow
    - Has inputs that match the system_input
    - Has outputs that match the system_input (just pass through)
    - Does not have an agent
    - Does not have a task
    - Does not have listen elements
    Example:
    ```xml
    <event>
        <name>on_start</name>
        <inputs>
            <input>
                <key>user_topic</key>
                <description>The user's topic that user wants to write a
                ↪ wikipiead-like article about.</description>
            </input>
        </inputs>
        <outputs>
            <output>
                <key>user_topic</key>
```

```xml
                <description>The user's topic that user wants to write a
                ↪ wikipiead-like article about.</description>
                <action>
                    <type>RESULT</type>
                </action>
            </output>
        </outputs>
    </event>
    ```

1. For simple sequential flows:
   - Use single output with RESULT type
   - No condition is needed
   - Next event in chain listening to this event will be triggered
   ↪ automatically

2. For conditional flows:
   - Multiple outputs must each have a condition
   - Conditions should be mutually exclusive
   - Each output should specify appropriate action type
   - `GOTO` action should have a value which is the name of the event
   ↪ to go to

3. Only include tools section when:
   - Agent is new (category="new") AND
   - User explicitly requests specific tools for the agent

4. Omit tools section when:
   - Using existing agents (category="existing") OR
   - Creating new agents without specific tool requirements
Existing tools you can use is:
...

Existing agents you can use is:
...

The name of existing workflows: [...]. The name of the new workflow
↪ you are creating should be DIFFERENT from these names according to
↪ the speciality of the workflow.

COMMON WORKFLOW PATTERNS:

1. If-Else Pattern (Conditional Branching):
```xml
<event>
    <name>analyze_data</name>
    <task>Analyze the data and determine next steps</task>
    <outputs>
        <output>
            <key>positive_case</key>
            <description>Handle positive case</description>
            <condition>If data meets criteria A</condition>
            <action>
                <type>RESULT</type>
            </action>
        </output>
        <output>
            <key>negative_case</key>
            <description>Handle the negative case</description>
            <condition>If data does not meet criteria A</condition>
            <action>
                <type>ABORT</type>
            </action>
        </output>
    </outputs>
```

```
</event>
```

2. Parallelization Pattern (Concurrent Execution):
```xml
<!-- Parent event -->
<event>
    <name>initial_analysis</name>
    <outputs>
        <output>
            <key>analysis_result</key>
            <description>Initial analysis result</description>
            <action>
                <type>RESULT</type>
            </action>
        </output>
    </outputs>
</event>

<!-- Multiple events listening to the same parent -->
<event>
    <name>technical_analysis</name>
    <listen>
        <event>initial_analysis</event>
    </listen>
    <outputs>
        <output>
            <key>technical_result</key>
            <description>Technical analysis result</description>
            <action>
                <type>RESULT</type>
            </action>
        </output>
    </outputs>
</event>

<event>
    <name>financial_analysis</name>
    <listen>
        <event>initial_analysis</event>
    </listen>
    <outputs>
        <output>
            <key>financial_result</key>
            <description>Financial analysis result</description>
            <action>
                <type>RESULT</type>
            </action>
        </output>
    </outputs>
</event>

<!-- Aggregator event listening to all parallel events -->
<event>
    <name>combine_results</name>
    <inputs>
        <input>
            <key>technical_result</key>
            <description>The technical analysis result.</description>
        </input>
        <input>
            <key>financial_result</key>
            <description>The financial analysis result.</description>
        </input>
    </inputs>
```

```
   <listen>
      <event>technical_analysis</event>
      <event>financial_analysis</event>
   </listen>
   <!-- This event will only execute when ALL listened events
   ↪ complete -->
</event>
```

3. Evaluator-Optimizer Pattern (Iterative Refinement):
```xml
<event>
   <name>generate_content</name>
   <outputs>
      <output>
         <key>content</key>
         <description>Generated content</description>
         <action>
            <type>RESULT</type>
         </action>
      </output>
   </outputs>
</event>

<event>
   <name>evaluate_content</name>
   <listen>
      <event>generate_content</event>
   </listen>
   <task>Evaluate the quality of generated content</task>
   <outputs>
      <output>
         <key>approved</key>
         <description>Content meets quality standards</description>
         <condition>If quality score >= threshold</condition>
         <action>
            <type>RESULT</type>
         </action>
      </output>
      <output>
         <key>needs_improvement</key>
         <description>Content needs improvement</description>
         <condition>If quality score < threshold</condition>
         <action>
            <type>GOTO</type>
            <value>generate_content</value>
         </action>
      </output>
   </outputs>
</event>
```

IMPORTANT NOTES ON PATTERNS:
0. The above patterns are incomplete which some mandatory elements
↪ are missing due to the limitation of context length. In real-world,
↪ you could refer to the logic of the patterns to create a complete
↪ and correct workflow.

1. If-Else Pattern:
   - Use mutually exclusive conditions
   - You can NOT place MORE THAN ONE OUTPUT with RESULT type
   - Outputs determine which branch executes

2. Parallelization Pattern:
   - Multiple events can listen to the same parent event
```

```
  - Aggregator event must list ALL parallel events in its listen
↪ section
  - All parallel events must complete before aggregator executes
  - Model of agents in every parallel event could be different

3. Evaluator-Optimizer Pattern:
  - Use GOTO action for iteration
  - Include clear evaluation criteria in conditions
  - Have both success and retry paths
  - Consider adding maximum iteration limit in global_variables
EXAMPLE:

User: I want to build a workflow that can help me to write a
↪ wikipiead-like article about the user's topic. It should:
1. Search the web for the user's topic.
2. Write an outline for the user's topic.
3. Evaluate the outline. If the outline is not good enough, repeat
↪ the outline step, otherwise, continue to write the article.
4. Write the article.

The form should be:
<workflow>
    <name>wiki_article_workflow</name>
    <system_input>
        <key>user_topic</key>
        <description>The user's topic that user wants to write a
        ↪ wikipiead-like article about.</description>
    </system_input>
    <system_output>
        <key>article</key>
        <description>The article that satisfies the user's
        ↪ request.</description>
    </system_output>
    <agents>
        <agent category="existing">
            <name>Web Surfer Agent</name>
            <description>This agent is used to search the web for the
            ↪ user's topic.</description>
        </agent>
        <agent category="new">
            <name>Outline Agent</name>
            <description>This agent is used to write an outline for the
            ↪ user's topic.</description>
        </agent>
        <agent category="new">
            <name>Evaluator Agent</name>
            <description>This agent is used to evaluate the outline of
            ↪ the user's topic.</description>
        </agent>
        <agent category="new">
            <name>Article Writer Agent</name>
            <description>This agent is used to write the article for the
            ↪ user's topic.</description>
        </agent>
    </agents>

    <events>
        <event>
            <name>on_start</name>
            <inputs>
                <input>
                    <key>user_topic</key>
                    <description>The user's topic that user wants to write
                    ↪ a wikipiead-like article about.</description>
                </input>
```

```
1782            </inputs>
1783            <outputs>
1784                <output>
1785                    <key>user_topic</key>
1786                    <description>The user's topic that user wants to write
                     ↪ a wikipiead-like article about.</description>
1787                    <action>
1788                        <type>RESULT</type>
1789                    </action>
1790                </output>
1791            </outputs>
1792        </event>
1793        <event>
1794            <name>on_search</name>
1795            <inputs>
1796                <input>
1797                    <key>user_topic</key>
1798                    <description>The user's topic that user wants to write
                     ↪ a wikipiead-like article about.</description>
1799                </input>
1800            </inputs>
1801            <task>
1802                search the information about the topic and return the
                 ↪ result.
1803            </task>
1804            <outputs>
1805                <output>
1806                    <key>search_result</key>
1807                    <description>The search result of the user's
                     ↪ topic.</description>
1808                    <action>
1809                        <type>RESULT</type>
1810                    </action>
1811                </output>
1812            </outputs>
1813            <listen>
1814                <event>on_start</event>
1815            </listen>
1816            <agent>
1817                <name>Web Surfer Agent</name>
1818                <model>claude-3-5-sonnet-20241022</model>
            </agent>
        </event>
        <event>
            <name>on_outline</name>
            <inputs>
                <input>
                    <key>search_result</key>
                    <description>The search result of the user's
                     ↪ topic.</description>
                </input>
            </inputs>
            <task>
                write an outline for the user's topic.
            </task>
            <outputs>
                <output>
                    <key>outline</key>
                    <description>The outline of the user's
                     ↪ topic.</description>
                    <action>
                        <type>RESULT</type>
                    </action>
                </output>
            </outputs>
```

```xml
                    <listen>
                        <event>on_start</event>
                    </listen>
                    <agent>
                        <name>Outline Agent</name>
                        <model>claude-3-5-sonnet-20241022</model>
                    </agent>
                </event>
                <event>
                    <name>on_evaluate</name>
                    <inputs>
                        <input>
                            <key>outline</key>
                            <description>The outline of the user's
                            ↪ topic.</description>
                        </input>
                    </inputs>
                    <task>
                        evaluate the outline of the user's topic.
                    </task>
                    <outputs>
                        <output>
                            <key>positive_feedback</key>
                            <description>The positive feedback of the outline of
                            ↪ the user's topic.</description>
                            <condition>
                                If the outline is good enough, give positive
                                ↪ feedback.
                            </condition>
                            <action>
                                <type>RESULT</type>
                            </action>
                        </output>
                        <output>
                            <key>negative_feedback</key>
                            <description>The negative feedback of the outline of
                            ↪ the user's topic.</description>
                            <condition>
                                If the outline is not good enough, give negative
                                ↪ feedback.
                            </condition>
                            <action>
                                <type>GOTO</type>
                                <value>on_outline</value>
                            </action>
                        </output>
                    </outputs>
                    <listen>
                        <event>on_outline</event>
                    </listen>
                    <agent>
                        <name>Evaluator Agent</name>
                        <model>claude-3-5-sonnet-20241022</model>
                    </agent>
                </event>
                <event>
                    <name>on_write</name>
                    <inputs>
                        <input>
                            <key>outline</key>
                            <description>The outline of user's topic.</description>
                        </input>
                    </inputs>
                    <task>
                        write the article for the user's topic.
```

```
          </task>
          <outputs>
              <output>
                  <key>article</key>
                  <description>The article of the user's
                  ↪ topic.</description>
                  <action>
                      <type>RESULT</type>
                  </action>
              </output>
          </outputs>
          <listen>
              <event>on_evaluate</event>
          </listen>
          <agent>
              <name>Article Writer Agent</name>
              <model>claude-3-5-sonnet-20241022</model>
          </agent>
      </event>
  </events>
</workflow>

GUIDELINES:
1. Each event should have clear inputs and outputs
2. Use conditions to handle different outcomes
3. Properly chain events using the listen element
4. Review steps should be included for quality control
5. Action types should be either RESULT or ABORT

Follow these examples and guidelines to create appropriate workflow
↪ forms based on user requirements.
```

Listing 15: Tools of **Workflow Editor Agent**

```
[list_agents, create_agent, execute_command, read_agent,
terminal_page_down, terminal_page_up, terminal_page_to,
list_workflows, create_workflow, run_workflow]
```

Listing 16: System Prompt of **Workflow Editor Agent**

```
You are a Workflow Creator specialized in the MetaChain framework.
↪ Your primary responsibility is to create and manage workflows based
↪ on XML-formatted workflow forms.

CORE RESPONSIBILITIES:
1. Parse and implement workflow forms
2. Create necessary agents if specified in the workflow
3. Create and manage workflows
4. Execute workflows as needed

AVAILABLE FUNCTIONS:
1. Workflow Management:
   - 'create_workflow': Create new workflows based on the workflow form
   - 'run_workflow': Execute the created workflow
   - 'list_workflows': Display all available workflows

2. Agent Management (when needed):
   - 'create_agent': Create new agents if specified in the workflow
   ↪ form. If no tools are explicitly specified, use empty tool list
   ↪ ([])
   - 'read_agent': Retrieve existing agent definitions before updates
   - 'list_agents': Display all available agents
```

```
3. System Tools:
  - `execute_command`: Handle system dependencies
  - `terminal_page_down`, `terminal_page_up`, `terminal_page_to`:
  ↪ Navigate terminal output

WORKFLOW CREATION PROCESS:

1. Parse Workflow Form:
  - Analyze the workflow form carefully
  - Identify any new agents that need to be created
  - Understand the workflow structure and requirements

2. Create Required Agents:
  - For each new agent in the workflow form:
    * Use `create_agent` with appropriate parameters
    * If no tools specified, use empty tool list ([])
    * Verify agent creation success

3. Create Workflow:
  - Use `create_workflow` to generate the workflow
  - Ensure all required agents exist
  - Validate workflow structure

4. Execute Workflow:
  - Use `run_workflow` to execute the created workflow
  - Monitor execution progress
  - Handle any errors appropriately

BEST PRACTICES:
1. Always check if required agents exist before creating new ones
2. Use empty tool list ([]) when no specific tools are mentioned
3. Validate workflow creation before execution
4. Follow the exact specifications from the workflow form XML
5. Handle errors and dependencies appropriately

Remember: Your primary goal is to create and execute workflows
↪ according to the provided workflow forms, creating any necessary
↪ agents along the way.
```

## A.7 SUPPLEMENTARY EXPERIMENTAL FINDINGS

### A.7.1 CASE OF 'DAVINCI AGENT'

**Task with Single Agent**. AutoAgent can create tools for third-party APIs (RapidAPI, Hugging Face). We demonstrated this by generating a DaVinci Agent for image creation and refinement. This shows AutoAgentś capability to build task-specific agents from natural language.

```
I want to create a `DaVinci Agent' that can help me to generate the
 image with natural language. it can:
1. generate the image with natural language and save it to the specified
 path on the local machine using the HF model 'Sana_600M_1024px_diffusers'
2. evaluate the image using `visual_question_answering` tool according
 to the given image.
3. interatively refine generated image based on the evaluation result.
```

**Automated Agent Creation and Execution**. AutoAgent begins generating an XML table from the natural language requirements, using existing tools and agents. This structured form is then passed to the Tool Editor Agent, which creates the necessary generate_image and refine_image tools. The Agent Editor Agent composes the DaVinci Agent by integrating the new tools with an existing visual_question_answering tool. This agent is executed, generating and storing several logo designs, as shown in Fig 3. Due to limited local resources, a smaller model was used, yet the agent successfully completed the task. This demonstrates AutoAgent's seamless creation of the tailored agent for complex, open-ended design challenges.

The XML Form of **DaVinci Agent** generated by the Agent Profiling Agent is shown in List 17. The logos of our AutoAgent generated by the created DaVinci Agent are displayed in Fig 3.

Listing 17: AI-generated Creation Profile of **DaVinci Agent**

```xml
<agents>
   <system_input>
      A natural language description for generating an image and
       evaluating its quality.
   </system_input>
   <system_output>
      <key>image_evaluation</key>
      <description>The evaluation of the generated image after
       processing.</description>
   </system_output>
   <agent>
      <name>DaVinci Agent</name>
      <description>The DaVinci Agent is designed to generate images
       from natural language
descriptions, evaluate them using predefined criteria, and
 iteratively refine the image
based on the evaluations.</description>
      <instructions>Use the HF model
       'Efficient-Large-Model/Sana_600M_1024px_diffusers' to
generate images from provided descriptions, evaluate these using
 visual QA, and refine based
on feedback.</instructions>
      <tools category="existing">
         <tool>
            <name>visual_question_answering</name>
            <description>This tool is used to answer questions about
             attached images or
videos.</description>
         </tool>
      </tools>
      <tools category="new">
         <tool>
            <name>generate_image</name>
            <description>Generate an image from a natural language
             description and save
it to a specified path using the HF model
'Efficient-Large-Model/Sana_600M_1024px_diffusers'.</description>
         </tool>
         <tool>
            <name>refine_image</name>
            <description>Make iterative adjustments to the generated
             image based on
evaluation results to meet quality criteria.</description>
         </tool>
      </tools>
      <agent_input>
         <key>image_description</key>
         <description>A natural language description to generate an
          image.</description>
      </agent_input>
      <agent_output>
         <key>image_evaluation</key>
         <description>The evaluation of the generated image after
processing.</description>
      </agent_output>
   </agent>
</agents>
```

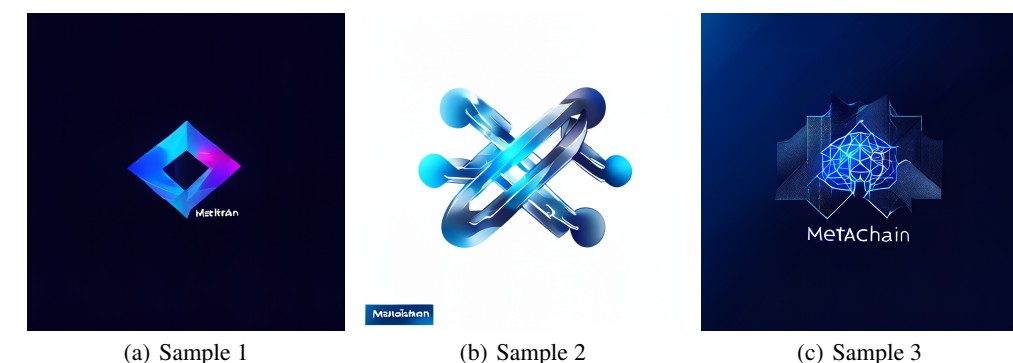

(a) Sample 1  (b) Sample 2  (c) Sample 3

Figure 3: The AI-generated AutoAgent's logos.

### A.7.2 CASE OF 'FINANCIAL AGENT'

The XML Form of **Financial Agent** generated by the Agent Profiling Agent is shown in List 18. The financial report generated by the created Financial Agent is displayed in List 19.

Listing 18: AI-generated Creation Profile of **Financial Agent**

```xml
<agents>
  <system_input>
    Financial management requests, including:
    1. Managing private financial documents stored in the
    'financial_docs' folder
    2. Retrieving online financial information for specific
    companies (balance sheets, cash flow statements, income
    statements)
  </system_input>
  <system_output>
    <key>financial_response</key>
    <description>Detailed response containing either document
    management results or requested financial
    information.</description>
  </system_output>
  <agent>
    <name>Document Manager Agent</name>
    <description>Specialized agent for managing and analyzing
    private financial documents stored locally.</description>
    <instructions>You are responsible for managing financial
    documents in the 'financial_docs' folder. Your tasks include:
1. Organizing and categorizing financial documents
2. Extracting relevant information from documents
3. Providing summaries and analyses of document contents
4. Maintaining document organization and searchability</instructions>
    <tools category="existing">
      <tool>
        <name>save_raw_docs_to_vector_db</name>
        <description>Save the financial documents to the vector
        database for efficient searching and
        retrieval.</description>
      </tool>
      <tool>
        <name>query_db</name>
        <description>Search through stored financial documents to
        find relevant information.</description>
      </tool>
      <tool>
        <name>visual_question_answering</name>
```

```
2106              <description>Process and analyze any financial charts,
2107                 graphs, or visual data in the documents.</description>
2108              </tool>
2109          </tools>
2110          <agent_input>
2111              <key>doc_request</key>
2112              <description>User request related to managing or querying
                  private financial documents.</description>
2113          </agent_input>
2114          <agent_output>
2115              <key>doc_response</key>
2116              <description>Results of document management operations or
                  requested document information.</description>
2117          </agent_output>
2118      </agent>
2119      <agent>
2120          <name>Market Research Agent</name>
2121          <description>Specialized agent for retrieving and analyzing
              online financial information for publicly traded
2122          companies.</description>
2123          <instructions>You are responsible for retrieving and analyzing
2124          financial information from online sources. Your tasks include:
2125  1. Fetching balance sheets, cash flow statements, and income
      statements
2126  2. Analyzing financial metrics and trends
2127  3. Providing clear summaries of financial data
2128  4. Ensuring data accuracy and proper citation of
2129   sources</instructions>
2130          <tools category="new">
2131              <tool>
2132                  <name>get_balance_sheet</name>
2133                  <description>Retrieve balance sheet data for a specific
                      ticker over a given time period.</description>
2134              </tool>
2135              <tool>
2136                  <name>get_cash_flow</name>
2137                  <description>Retrieve cash flow statement data for a
                      specific ticker over a given time period.</description>
2138              </tool>
2139              <tool>
2140                  <name>get_income_statement</name>
2141                  <description>Retrieve income statement data for a
                      specific ticker over a given time period.</description>
2142              </tool>
2143              <tool>
2144                  <name>analyze_financial_data</name>
2145                  <description>Analyze and summarize financial statements
                      to provide meaningful insights.</description>
2146              </tool>
2147          </tools>
2148          <agent_input>
2149              <key>market_request</key>
2150              <description>User request for online financial information
                  including ticker symbol and time period.</description>
2151          </agent_input>
2152          <agent_output>
2153              <key>market_response</key>
2154              <description>Requested financial information and analysis
                  from online sources.</description>
2155          </agent_output>
2156      </agent>
2157  </agents>
2158
2159
```

Listing 19: The report generated by AI-generated **Financial Agent**

```
Based on the comprehensive analysis of both historical performance
 from 10-K reports and current financial data, here is the final
 investment recommendation for 2025:

Investment Recommendation Summary for 2025:

1. Portfolio Allocation Strategy:
- Microsoft (MSFT): 60% of technology allocation
- Apple (AAPL): 40% of technology allocation

2. Investment Rationale:

Microsoft (MSFT) - Primary Position (60%):
- Leading AI integration and monetization potential
- Superior profit margins (37.61% vs 15.52%)
- Strong cloud revenue growth (23% YoY)
- Successful Activision Blizzard integration
- Robust enterprise market position
- Higher recent revenue growth momentum
- Strong cash position ($78.43B)
- Clear AI leadership advantages

Apple (AAPL) - Supporting Position (40%):
- Superior capital efficiency (ROC: 43.01%)
- Higher Return on Assets (21.24%)
- Strong services growth potential
- Reliable cash flow generation
- Consistent shareholder returns
- Valuable ecosystem lock-in
- Strong emerging market potential
- Value characteristics at current levels

3. Key Monitoring Points for 2025:

Microsoft:
- Cloud market share gains
- AI revenue monetization
- Gaming division integration
- Enterprise spending trends
- Margin sustainability
- Regulatory developments

Apple:
- Services revenue growth
- iPhone replacement cycles
- Emerging market penetration
- Margin pressure management
- AI/ML integration progress
- Supply chain diversification

4. Risk Management Strategy:

Short-term (0-6 months):
- Monitor AI monetization progress
- Track iPhone 15 cycle performance
- Watch macro economic indicators
- Observe cloud growth metrics

Medium-term (6-12 months):
- Evaluate competitive positions in AI
- Assess services growth trajectory
- Monitor regulatory environment
- Track enterprise spending trends
```

```
2214   Long-term (12+ months):
2215   - Evaluate ecosystem strength
2216   - Monitor technological shifts
2217   - Assess market share dynamics
2218   - Track return on capital trends
2219
2220   5. Investment Approach:
2221
2222   Implementation Strategy:
2223   - Consider dollar-cost averaging for new positions
2224   - Maintain quarterly rebalancing schedule
       - Set stop-loss levels based on risk tolerance
2225   - Monitor position sizes relative to overall portfolio
2226
2227   Adjustment Triggers:
2228   - Significant margin deterioration
2229   - Market share losses in key segments
       - Major regulatory actions
2230   - Substantial technological disruption
2231   - Material changes in competitive position
2232
2233   The 60/40 split in favor of Microsoft reflects:
2234   1. Stronger near-term growth prospects in AI and cloud
       2. Higher profit margins and improving efficiency metrics
2235   3. Better positioning in enterprise markets
2236   4. More diverse revenue streams
2237   5. Clear leadership in next-generation technologies
2238
       However, Apple's position remains crucial due to:
2239   1. Superior capital efficiency
2240   2. Strong ecosystem advantages
2241   3. Significant services growth potential
2242   4. Value characteristics at current levels
       5. Consistent shareholder returns
2243
2244   This balanced approach provides exposure to both immediate growth
2245    opportunities through Microsoft's AI and cloud leadership, while
        maintaining stability through Apple's efficient operations and
2246    ecosystem strength. Regular monitoring and rebalancing will be
2247    essential to maintain optimal positioning as market conditions evolve
2248    through 2025.
2249
2250
```

### A.7.3   CASE OF 'MAJORITY VOTING' WORKFLOW

The XML Form of **'Majority Voting' Workflow** generated by the Workflow Profiling Agent is shown in List 20. A comparison between the math reasoning results of the created 'Majority Voting' Workflow and the math reasoning results of a single DeepSeek-V3 model is presented in Tab 5.

Listing 20: AI-generated Creation Profile of **'Majority Voting' workflow**

```xml
<workflow>
   <name>parallel_math_solver_workflow</name>
   <system_input>
      <key>math_problem</key>
      <description>The math problem that needs to be
       solved.</description>
   </system_input>
   <system_output>
      <key>final_solution</key>
      <description>The final solution determined by majority
       voting.</description>
   </system_output>
   <agents>
      <agent category="new">
```

```
2268            <name>Math Solver Agent</name>
2269            <description>This agent solves mathematical problems using
2270             analytical and systematic approaches.</description>
2271        </agent>
2272        <agent category="new">
2273            <name>Vote Aggregator Agent</name>
2274            <description>This agent aggregates solutions from different
2275             solvers and determines the final answer through majority
2276             voting.</description>
2277        </agent>
2278    </agents>
2279    <events>
2280        <event>
2281            <name>on_start</name>
2282            <inputs>
2283                <input>
2284                    <key>math_problem</key>
2285                    <description>The math problem that needs to be
2286                     solved.</description>
2287                </input>
2288            </inputs>
2289            <outputs>
2290                <output>
2291                    <key>math_problem</key>
2292                    <description>The math problem that needs to be
2293                     solved.</description>
2294                    <action>
2295                        <type>RESULT</type>
2296                    </action>
2297                </output>
2298            </outputs>
2299        </event>
2300        <event>
2301            <name>solve_with_gpt4</name>
2302            <inputs>
2303                <input>
2304                    <key>math_problem</key>
2305                    <description>The math problem that needs to be
2306                     solved.</description>
2307                </input>
2308            </inputs>
2309            <task>Solve the math problem using systematic approach with
2310             GPT-4.</task>
2311            <outputs>
2312                <output>
2313                    <key>gpt4_solution</key>
2314                    <description>The solution from GPT-4
2315                     solver.</description>
2316                    <action>
2317                        <type>RESULT</type>
2318                    </action>
2319                </output>
2320            </outputs>
2321            <listen>
                    <event>on_start</event>
                </listen>
                <agent>
                    <name>Math Solver Agent</name>
                    <model>gpt-4o-2024-08-06</model>
                </agent>
            </event>
            <event>
                <name>solve_with_claude</name>
                <inputs>
                    <input>
```

```xml
                    <key>math_problem</key>
                    <description>The math problem that needs to be
                     solved.</description>
                </input>
            </inputs>
            <task>Solve the math problem using systematic approach with
             Claude.</task>
            <outputs>
                <output>
                    <key>claude_solution</key>
                    <description>The solution from Claude
                     solver.</description>
                    <action>
                        <type>RESULT</type>
                    </action>
                </output>
            </outputs>
            <listen>
                <event>on_start</event>
            </listen>
            <agent>
                <name>Math Solver Agent</name>
                <model>claude-3-5-sonnet-20241022</model>
            </agent>
        </event>
        <event>
            <name>solve_with_deepseek</name>
            <inputs>
                <input>
                    <key>math_problem</key>
                    <description>The math problem that needs to be
                     solved.</description>
                </input>
            </inputs>
            <task>Solve the math problem using systematic approach with
             DeepSeek.</task>
            <outputs>
                <output>
                    <key>deepseek_solution</key>
                    <description>The solution from DeepSeek
                     solver.</description>
                    <action>
                        <type>RESULT</type>
                    </action>
                </output>
            </outputs>
            <listen>
                <event>on_start</event>
            </listen>
            <agent>
                <name>Math Solver Agent</name>
                <model>deepseek/deepseek-chat</model>
            </agent>
        </event>
        <event>
            <name>aggregate_solutions</name>
            <inputs>
                <input>
                    <key>gpt4_solution</key>
                    <description>The solution from GPT-4
                     solver.</description>
                </input>
                <input>
                    <key>claude_solution</key>
```

```
                    <description>The solution from Claude
                     solver.</description>
                </input>
                <input>
                    <key>deepseek_solution</key>
                    <description>The solution from DeepSeek
                     solver.</description>
                </input>
            </inputs>
            <task>Compare all solutions and determine the final answer
             through majority voting.</task>
            <outputs>
                <output>
                    <key>final_solution</key>
                    <description>The final solution determined by majority
                     voting.</description>
                    <action>
                        <type>RESULT</type>
                    </action>
                </output>
            </outputs>
            <listen>
                <event>solve_with_gpt4</event>
                <event>solve_with_claude</event>
                <event>solve_with_deepseek</event>
            </listen>
            <agent>
                <name>Vote Aggregator Agent</name>
                <model>deepseek/deepseek-chat</model>
            </agent>
        </event>
    </events>
</workflow>
```

Table 5: Comparison of the single LLM (Deepseek-v3) and AI-generated Majority Voting workflow.

---

**Problem:** Simplify $\tan 100° + 4\sin 100°$.

---

**Ground Truth:** $-\sqrt{3}$

---

**AI-generated Majority Voting workflow:** Let me analyze all three solutions:

1. GPT-4 Solution: - Does not provide a definitive numerical answer - Focuses on theoretical approach and approximations - Inconclusive result

2. Claude Solution: - Uses supplementary angle properties - Converts 100° to 180° - 80° - Arrives at $-\sqrt{3}$ through numerical calculations - Final answer: $-\sqrt{3}$

3. Deepseek Solution: - Uses similar supplementary angle properties - Also converts 100° to 180° - 80° - Goes through detailed algebraic steps - Final answer: $\sqrt{3}$

Analysis: 1. Two solutions provide definitive answers ($-\sqrt{3}$ and $\sqrt{3}$), while one is inconclusive 2. Both Claude and Deepseek use similar approaches with supplementary angles 3. The main difference is in the sign of the final answer 4. Claude's solution provides more explicit numerical calculations 5. Looking at the steps: - 100° being in quadrant II - Using supplementary angle properties - tan 100° being negative in quadrant II - The final combination with 4sin 100°

The Claude solution appears more precise in tracking signs and considering quadrant properties, making its answer more reliable.

Final Answer: The final answer is $-\sqrt{3}$. I hope it is correct.

---

**Deepseek-v3:** To simplify $\tan 100° + 4\sin 100°$, we proceed as follows: 1. Recall that $\tan\theta = \frac{\sin\theta}{\cos\theta}$. Thus:

$$\tan 100° = \frac{\sin 100°}{\cos 100°}.$$

2. Substitute this into the original expression:

$$\tan 100° + 4\sin 100° = \frac{\sin 100°}{\cos 100°} + 4\sin 100°.$$

...

16. Therefore, the simplified form of $\tan 100° + 4\sin 100°$ is:

$$\boxed{\sqrt{3}}.$$

Final Answer: The final answer is $\sqrt{3}$ . I hope it is correct.

---

**Problem:** What integer $n$ satisfies $0 \le n < 18$ and

$$n \equiv -11213141 \pmod{18} ?$$

---

**Ground Truth:** 13

**AI-generated Majority Voting workflow:** Let me compare the three solutions:
1. GPT-4 Solution: Arrives at 13 through direct division and remainder calculation - Used division: 11213141 ÷ 18 - Found remainder of 5 for positive number - Adjusted for negative sign to get 13
2. Claude Solution: Arrives at 13 through division and remainder - Divided -11213141 by 18 - Found remainder directly - Got 13 as the result in the range [0,18)
3. Deepseek Solution: Arrives at 5 through : - Division and remainder calculation - Used divisibility rules for 2 and 9 - Got 5 as final answer
Analyzing the solutions: - Two solutions (GPT-4 and Claude) agree on 13 - One solution (Deepseek) gives

5 - Both GPT-4 and Claude show clearer arithmetic progressions - The key difference is in handling the negative sign in the modulo operation - When working with negative numbers in modulo arithmetic, we need to find the positive equivalent in the range [0,n) - The correct approach is to add the modulus (18) to the negative remainder until we get a number in the range [0,18)
By majority voting and verification: - 13 is supported by two solutions -
The reasoning in these solutions is more complete - We can verify: $-11213141 \equiv 13 \pmod{18}$ and $0 \le 13 < 18$
Final Answer: The final answer is 13. I hope it is correct.

**Deepseek-v3:** To solve for the integer $n$ that satisfies $0 \leq n < 18$ and $n \equiv -11213141 \pmod{18}$, we need to find the remainder when $-11213141$ is divided by 18. Here's the step-by-step solution:

1. **Simplify the modulus**: First, note that $-11213141 \pmod{18}$ is equivalent to finding the positive remainder when $-11213141$ is divided by 18. Since the modulus operation is periodic with period 18, we can add multiples of 18 to $-11213141$ until we obtain a positive number within the range $0 \leq n < 18$.

2. **Calculate the remainder**: To find $-11213141 \pmod{18}$, we can instead compute $11213141 \pmod{18}$ and then take the negative of that result, adjusting it to be positive if necessary.

- First, compute $11213141 \div 18$:

$$11213141 \div 18 = 623000 \text{ with a remainder of } 11213141 - 18 \times 623000 = 11213141 - 11214000 = -859$$

This means $11213141 \equiv -859 \pmod{18}$.

- Now, compute $-859 \pmod{18}$:

$$-859 \div 18 = -48 \text{ with a remainder of } -859 - 18 \times (-48) = -859 + 864 = 5$$

So, $-859 \equiv 5 \pmod{18}$.

- Therefore, $-11213141 \equiv 5 \pmod{18}$.

3. **Final answer**: The integer $n$ that satisfies $0 \leq n < 18$ and $n \equiv -11213141 \pmod{18}$ is 5.

Final Answer: The final answer is $5$. I hope it is correct.

---

**Problem:** In the diagram below, we have $\sin \angle RPQ = \frac{7}{25}$. What is $\cos \angle RPS$?

[asy]
pair R,P,Q,SS;
SS = (-2,0);
P = (0,0);
Q = (2,0);
R = rotate(aSin(7/25))*(1.5,0);
dot("$S$",SS,S);
dot("$Q$",Q,S);
dot("$R$",R,N);
dot("$P$",P,S);
draw(Q--SS);
draw(P--R);
[/asy]

---

**Ground Truth:** $-\frac{24}{25}$

---

**AI-generated Majority Voting workflow:** Let me analyze all three solutions:

1. GPT-4 Solution: Final answer = $\frac{24}{25}$ - Used right triangle approach - Found cosine of RPQ = 24/25 - Assumed RPS = RPQ

2. Claude Solution: Final answer = $-\frac{24}{25}$ - Used supplementary angles property - Found cos RPQ = 24/25 - Used the fact that cos(supplementary angles) are negatives of each other - Concluded cos RPS = -cos RPQ = -24/25

3. DeepSeek Solution: Final answer = $\frac{7}{25}$ - Used complementary angles property - Found cos RPQ = 24/25 - Incorrectly concluded cos RPS = sin RPQ

Analyzing the approaches: 1. Looking at the diagram, RPS and RPQ are supplementary angles (they form a straight line) 2. For supplementary angles, their cosines are negatives of each other 3. Claude's solution correctly uses this property 4. GPT-4 incorrectly assumed the angles were equal 5. DeepSeek incorrectly used complementary angle properties

Claude's solution is the most mathematically sound because: - It correctly identifies that RPS and RPQ are supplementary angles - It properly applies the relationship between cosines of supplementary angles - The negative sign in the answer makes geometric sense given the angle's position

Final Answer: The final answer is $-\frac{24}{25}$. I hope it is correct.

---

**Deepseek-v3:** We are given that $\sin \angle RPQ = \frac{7}{25}$. Since $\angle RPQ$ and $\angle RPS$ are complementary angles (they form a right angle together), we have:

$$\cos \angle RPS = \sin \angle RPQ = \frac{7}{25}.$$

Final Answer: The final answer is $\boxed{\frac{7}{25}}$ . I hope it is correct.

---

**Problem:** In the circle with center $Q$, radii $AQ$ and $BQ$ form a right angle. The two smaller regions are tangent semicircles, as shown. The radius of the circle with center $Q$ is 14 inches. What is the radius of the smaller semicircle? Express your answer as a common fraction.

**Ground Truth:** $\frac{14}{3}$

**AI-generated Majority Voting workflow:** Let me analyze all three solutions:

1. GPT-4's solution: r = 7  2. Claude's solution: r = 14/3  3. Deepseek's solution: $r = 14(\sqrt{2} - 1)$

Each solution uses different approaches: - GPT-4 assumes the semicircles are equal and uses a simple division (14/2 = 7) - Claude uses the relationship r + r + r = R, leading to 14/3 - Deepseek uses the Pythagorean theorem and complex algebra, arriving at $14(\sqrt{2} - 1)$

Let's verify which one makes more sense:

1. Key geometric principles suggest that when two circles are tangent to each other and to two perpendicular lines (in this case, the radii forming the right angle), there's a special relationship where r + r + r = R, where r is the radius of the smaller circles and R is the radius of the larger circle.

2. This principle is well-established in geometry. When two circles are placed in a right angle and are tangent to each other and the lines forming the right angle, their radii follow this 3:1 ratio.

3. Therefore, Claude's solution using 3r = 14 leading to r = 14/3 is the most mathematically sound .

4. We can verify this: - It satisfies the tangency conditions - The semicircles fit perfectly in the right angle - The relationship maintains the proper geometric proportions

The GPT-4 solution (r = 7) is too large as it would make the semicircles overlap. The Deepseek solution $14(\sqrt{2} - 1)$ is more complex than necessary and doesn't align with the fundamental geometric principle for this configuration.

Final Answer: The final answer is $\frac{14}{3}$. I hope it is correct.

---

**Deepseek-v3:** Let's solve the problem step by step.

1. Understand the setup: We have a circle with center $Q$ and radius 14 inches. Radii $AQ$ and $BQ$ form a right angle, meaning the angle $\angle AQB = 90°$. There are two smaller semicircles tangent to the larger circle and to each other.

2. Key observations: - The two semicircles are tangent to the larger circle and to each other. This means the distance from the center $Q$ to the center of each semicircle is $14 - r$, where $r$ is the radius of the smaller semicircle. - The centers of the two semicircles and the center $Q$ form a right triangle because $\angle AQB = 90°$.

. . .

5. Simplify the answer: - The radius of the smaller semicircle is $14(\sqrt{2} - 1)$. However, this can be expressed as a common fraction:

$$r = 14(\sqrt{2} - 1) = 14\sqrt{2} - 14.$$

$$r = \frac{14\sqrt{2} - 14}{1}.$$

$$r = \frac{14(\sqrt{2} - 1)}{1}.$$

$$r = 14(\sqrt{2} - 1).$$

6. Final Answer: The radius of the smaller semicircle is $14(\sqrt{2} - 1)$ .

Final Answer: The final answer is $14(\sqrt{2} - 1)$. I hope it is correct.

---

A.7.4    DETAILED TRAJECTORY OF CREATING 'FINANCIAL AGENT'

To provide a more comprehensive demonstration of how AutoAgent receives user requirements and automates the creation of tools and agents, we present the specific agent trajectory for the creation of the Financial Agent as follows.

Table 6: Detailed Trajectory of Creating 'Financial Agent'.

| Role | Action | Content |
|---|---|---|
| ♟: **User** | Input | I want to create **'Financial** ↪ **Agent'** that can help me to ↪ do two kinds of tasks: 1. Manage the private ↪ financial docs. I have a ↪ folder that contain the ↪ financial docs in my local ↪ machine, and I want to help ↪ me to manage them. 2. Search the financial ↪ information online. You may ↪ help me to: - get balance sheets for a ↪ given ticker over a given ↪ period. - get cash flow statements for ↪ a given ticker over a given ↪ period. - get income statements for a ↪ given ticker over a given ↪ period.

Directly output the form in ↪ the XML format without ANY ↪ other text. |
| 🤖: **Agent Profiling Agent** | output | `<agents>`
    `<system_input>`
        Financial management
        ↪ requests, including:
        ...
    `</system_input>`
    `<system_output>`
        `<key>`financial_response`</key>`
        ...
    `</system_output>`
    `<agent>`
        `<name>`Document Manager
        ↪ Agent`</name>`
        ...
    `</agent>`
    `<agent>`
        `<name>`Market Research
        ↪ Agent`</name>`
        ...
    `</agent>`
`</agents>` |

| 🤖: **Tool Editor Agent** | `list_tools()` | <pre>{
    "save_raw_docs_to_vector_db":
    ↪ {
      "name":
      ↪ "save_raw_docs_to_vector_db",
      "func_name":
      ↪ "save_raw_docs_to_vector_db",
      "args": [
        "context_variables",
        "doc_name",
        "saved_vector_db_name",
        "overwrite"
      ],
      "docstring":
      ...
    },
    ...
}</pre> |
|---|---|---|
| 🤖: **Tool Editor Agent** | `get_api_plugin_tools_doc ('query_text'='API for`
↪ `getting financial statements (balance`
↪ `sheet, cash flow,`
↪ `income statement) from stock market')` | The rank 1 referenced tool
↪ documentation is:
API Name: Company Income
↪ Statement
...
The rank 2 referenced tool
↪ documentation is:
...
... |
| 🤖: **Tool Editor Agent** | `create_tool(`
`'tool_name'=`
`'get_balance_sheet',`
`'tool_code'=`
`'from metachain.types`
↪ `import Result, Agent`
`...`
`')` | [SUCCESS] Successfully created
↪ tool: get_balance_sheet in
↪ /workplace/metachain/metachain/
tools/get_balance_sheet.py |
| 🤖: **Tool Editor Agent** | `run_tool(`
`'tool_name'=`
`'get_balance_sheet',`
`'run_code'=`
`'from metachain.tools`
↪ `import`
↪ `get_balance_sheet`
`result =`
`get_balance_sheet("AAPL",`
↪ `"QUARTERLY")`
`print(result)')` | [The output of the tool
↪ 'run_tool' showing in the
↪ interactive terminal]
Terminal viewport position:
↪ Showing page 1 of 1.
=====================
[SUCCESS]
{
  "symbol": "AAPL:NASDAQ",
  "type": "stock",
  "period": "QUARTERLY",
  "balance_sheet": [
  ...
===================== |

| | | |
|---|---|---|
| 📱: **Tool Editor Agent** | ```create_tool(
'tool_name'=
'get_cash_flow',
'tool_code'=
'from metachain.types
↪ import Result, Agent
...
')``` | ```[SUCCESS] Successfully created
↪ tool: get_cash_flow in
↪ /workplace/metachain/metachain/
tools/get_cash_flow.py``` |
| 📱: **Tool Editor Agent** | ```run_tool(
'tool_name'=
'get_cash_flow',
'run_code'=
'from metachain.tools
↪ import get_cash_flow
result =
↪ get_cash_flow("AAPL",
↪ "QUARTERLY")
print(result)')``` | ```[The output of the tool
↪ `run_tool` showing in the
↪ interactive terminal]
Terminal viewport position:
↪ Showing page 1 of 1.
=======================
[SUCCESS]
{
  "symbol": "AAPL:NASDAQ",
  "type": "stock",
  "period": "QUARTERLY",
  "cash_flow": [
  ...
=======================``` |
| 📱: **Tool Editor Agent** | ```create_tool(
'tool_name'=
'get_income_statement',
'tool_code'=
'from metachain.types
↪ import Result, Agent
...')``` | ```[SUCCESS] Successfully created
↪ tool: get_income_statement
↪ in
↪ /workplace/metachain/metachain/
tools/get_income_statement.py``` |
| 📱: **Tool Editor Agent** | ```run_tool(
'tool_name'=
'get_income_statement',
'run_code'='from
↪ metachain.tools
↪ import
↪ get_income_statement
result =
↪ get_income_statement
("AAPL", "QUARTERLY")
print(result)')``` | ```[The output of the tool
↪ `run_tool` showing in the
↪ interactive terminal]
Terminal viewport position:
↪ Showing page 1 of 1.
=======================
[SUCCESS]
{
  "symbol": "AAPL:NASDAQ",
  "type": "stock",
  "period": "QUARTERLY",
  "income_statement": [
  ...
=======================``` |
| 📱: **Tool Editor Agent** | ```create_tool(
'tool_name'=
'analyze_financial_data',
'tool_code'=
'from metachain.types
↪ import Result, Agent
...
')``` | ```[SUCCESS] Successfully created
↪ tool:
↪ analyze_financial_data in
↪ /workplace/metachain/metachain/
tools/analyze_financial_data.py``` |

| | | |
|---|---|---|
| 2754
2755
2756
2757
2758
2759
2760
2761
2762
2763
2764
2765
2766
2767
2768
2769
2770
2771
2772
2773
2774 | **: Tool Editor Agent** | run_tool(
'tool_name'=
'analyze_financial_data',
'run_code'=
'from metachain.tools
↪ import
↪ get_income_statement,
↪ analyze_financial_data
income_data =
↪ get_income_statement
("AAPL", "QUARTERLY")
result =
↪ analyze_financial_data
(income_data)
print(result)') | [The output of the tool
↪ 'run_tool' showing in the
↪ interactive terminal]
Terminal viewport position:
↪ Showing page 1 of 1.
=======================
[SUCCESS]
Financial Analysis for
↪ AAPL:NASDAQ
----------------------
Income Statement Analysis:
For the period ending
↪ 2024-9-28:
...

Profitability Metrics:
...

Period-over-Period Comparison:
...

======================= |
| 2775
2776
2777
2778
2779
2780
2781
2782
2783
2784
2785
2786
2787
2788
2789
2790
2791
2792
2793
2794
2795 | **: Tool Editor Agent** | Output | Successfully created and
↪ tested all four financial
↪ tools:

1. get_balance_sheet – Created
↪ and tested successfully
↪ with AAPL stock data
2. get_cash_flow – Created and
↪ tested successfully with
↪ AAPL stock data
3. get_income_statement –
↪ Created and tested
↪ successfully with AAPL
↪ stock data
4. analyze_financial_data –
↪ Created and tested
↪ successfully by analyzing
↪ AAPL's income statement data

All tools are functioning as
↪ expected, providing
↪ financial data and analysis
↪ in proper JSON format. |

2796
2797
2798
2799
2800
2801
2802
2803
2804
2805
2806
2807

| 🤖: Agent Editor Agent | create_agent(
'agent_name'=
'Document Manager
↪ Agent',
'agent_description'=
'Specialized agent for
↪ managing and
↪ analyzing private
↪ financial documents
↪ stored locally.',
'agent_tools'=
['query_db', ...],
'agent_instructions'='You
↪ are a specialized
↪ financial document
↪ manager responsible
↪ for managing and
↪ analyzing financial
↪ documents stored in
↪ specified
↪ directories. Your
↪ tasks include:...') | Successfully created agent:
↪ Document Manager Agent in
↪ /workplace/metachain/metachain/
agents/document_manager_agent.py |
| --- | --- | --- |
| 🤖: Agent Editor Agent | create_agent(
'agent_name'=
'Market Research
↪ Agent',
'agent_description'=
'Specialized agent for
↪ retrieving and
↪ analyzing online
↪ financial
↪ information for
↪ publicly traded
↪ companies.',
'agent_tools'=
['get_cash_flow', ...],
'agent_instructions'=
'You are a specialized
↪ market research
↪ agent responsible
↪ for retrieving and
↪ analyzing financial
↪ information from
↪ online sources. Your
↪ primary
↪ responsibilities
↪ include: ...') | Successfully created agent:
↪ Market Research Agent in
↪ /workplace/metachain/metachain/
agents/market_research_agent.py |

| | | |
|---|---|---|
| 🤖: **Agent Editor Agent** | `create_orchestrator_agent('agent_name'='Financial Analysis ↪ Orchestrator', 'agent_description'='Orchestrator agent ↪ that coordinates ↪ between document ↪ management and ↪ market research to ↪ provide ↪ comprehensive ↪ financial analysis.', 'sub_agents'=[{'name': ↪ 'Document Manager ↪ Agent', ↪ 'agent_input': ↪ 'Document analysis request ...', 'agent_output': ↪ 'Analyzed information ...'}, {'name': ↪ 'Market Research ↪ Agent', ↪ 'agent_input': ↪ 'Market research request ...', 'agent_output': ↪ 'Market analysis ...'}], 'agent_instructions'='You are a Financial ↪ Analysis ↪ Orchestrator ↪ responsible for ↪ coordinating ↪ comprehensive ↪ financial analysis ↪ by leveraging both ↪ document analysis ↪ and market research ↪ capabilities. Your ↪ workflow: ...')` | `[ERROR] Failed to create ↪ agent. Error: File ↪ "/workplace/metachain/metachain/agents/financial_analysis_orchestrator.py", ↪ line 32     def     ↪ transfer_to_document_manager_agent(Document     ↪ analysis request     ↪ including directory     ↪ paths and analysis     ↪ requirements: str):                            ^^^^^^^^ SyntaxError: invalid syntax` |

| 🤖: Agent Editor Agent | create_orchestrator_agent ('agent_name'= 'Financial Analysis ↪ Orchestrator', 'agent_description'= 'Orchestrator agent ↪ that coordinates ↪ between document ↪ management and ↪ market research to ↪ provide ↪ comprehensive ↪ financial analysis.', 'sub_agents'=[{'name': ↪ 'Document Manager ↪ Agent', ↪ 'agent_input': ↪ 'doc_request', ↪ 'agent_output': ↪ 'doc_response'}, ↪ {'name': 'Market ↪ Research Agent', ↪ 'agent_input': ↪ 'market_request', ↪ 'agent_output': ↪ 'market_response'}], 'agent_instructions'= 'You are a Financial ↪ Analysis ↪ Orchestrator ↪ responsible for ↪ coordinating ↪ comprehensive ↪ financial analysis ↪ by leveraging both ↪ document analysis ↪ and market research ↪ capabilities. Your ↪ workflow: ...') | Successfully created agent: ↪ Financial Analysis ↪ Orchestrator in ↪ /workplace/metachain/metachain/ agents/financial_analysis_orchestrator.py |

🤖: **Agent Editor Agent**

```
run_agent(
'agent_name'=
'Financial Analysis
↪ Orchestrator',
'query'=
"Based␣on␣the␣10-K␣
↪ reports␣of␣AAPL␣and␣
↪ MSFT␣from␣the␣past␣5␣
↪ years␣in␣the␣docs␣
↪ directory␣'docs/
aapl-2020-2024-10K/'␣
↪ and␣'docs/
msft-2020-2024-10K/',␣
↪ along␣with␣AAPL's␣
↪ other␣reports␣'docs/
aapl-other-report/'␣
↪ and␣available␣data,␣
↪ conduct␣a␣
↪ comprehensive␣
↪ horizontal␣
↪ comparison,␣create␣a␣
↪ comparative␣analysis␣
↪ report,␣and␣provide␣
↪ constructive␣
↪ investment␣advice␣
↪ for␣investing␣in␣
↪ them␣in␣2025.",
'ctx_vars'='{}')
```

```
[The output of the tool
↪ 'run_agent' showing in the
↪ interactive terminal]
Terminal viewport position:
↪ Showing page 1 of 35.
[NOTE] The output of the tool
'run_agent', you can use
'terminal_page_up' to scroll up
and 'terminal_page_down' to
scroll down.  If there are many
pages with meaningless content
like progress bar or output of
generating directory structure
when there are many datasets in
the directory, you can use
'terminal_page_to' to move the
viewport to the end of terminal
where the meaningful content
is.
=======================
[SUCCESS]
**** Receive Task ****
Receiveing the task:
Based on the 10-K reports of
↪ AAPL and MSFT from the past
↪ 5 years in the docs
directory ...
**** Assistant Message ****
Financial Analysis
↪ Orchestrator: I'll help
↪ coordinate a comprehensive
↪ comparative analysis
↪ between Apple (AAPL)
and Microsoft (MSFT) using
↪ both historical document
↪ analysis and current market
research. Let's break this
↪ down into parallel
↪ workflows to gather all
↪ necessary
information.

1. First, let's analyze the
↪ historical documents (10-K
↪ reports and other
reports):
**** Tool Calls ****
[2025-01-22 05:58:48]
transfer_to_document_manager_agent
('doc_request'='...')
**** Tool Execution ****
[2025-01-22 05:58:48]
Tool Execution:
↪ transfer_to_document_manager_agent
Result:
---
Please analyze the following
↪ documents:
...

**** Assistant Message ****
Document Manager Agent: ...
=======================
```

| 🤖: **Agent Editor Agent** | terminal_page_to ('page_idx'=35) | [The output of the tool ↪ `run_agent` showing in the ↪ interactive terminal] Terminal viewport position: ↪ Showing page 35 of 35. [NOTE] The output of the tool `run_agent`, you can use `terminal_page_up` to scroll up and `terminal_page_down` to scroll down.  If there are many pages with meaningless content like progress bar or output of generating directory structure when there are many datasets in the directory, you can use `terminal_page_to` to move the viewport to the end of terminal where the meaningful content is. ====================== **** Assistant Message **** Financial Analysis ↪ Orchestrator: Based on the ↪ comprehensive analysis of ↪ both historical performance ↪ from 10-K reports and current financial ↪ data, here is the final ↪ investment recommendation for 2025:  Investment Recommendation ↪ Summary for 2025:  1. Portfolio Allocation ↪ Strategy: - Microsoft (MSFT): 60% of ↪ technology allocation - Apple (AAPL): 40% of ↪ technology allocation  2. Investment Rationale:  Microsoft (MSFT) - Primary ↪ Position (60%): ... ====================== |

| 🤖: **Agent Editor Agent** | Output | I have successfully created ↪ and tested the following ↪ agents:

1. Document Manager Agent – ↪ for managing and analyzing ↪ private financial documents
2. Market Research Agent – for ↪ retrieving and analyzing ↪ online financial information
3. Financial Analysis ↪ Orchestrator – coordinates ↪ between the two agents for ↪ comprehensive analysis

The agents were tested with a ↪ complex task of analyzing ↪ AAPL and MSFT's financial ↪ documents and providing ↪ investment advice. The ↪ orchestrator successfully:
1. Processed 10-K reports from ↪ both companies
2. Analyzed financial metrics ↪ and trends
3. Generated a comprehensive ↪ investment recommendation ↪ with:
  - Portfolio allocation ↪ strategy
  - Investment rationale for ↪ each company
  - Key monitoring points
  - Risk management strategy
  - Implementation approach

The test demonstrated that the ↪ agents can effectively work ↪ together to provide ↪ detailed financial analysis ↪ and actionable investment ↪ advice. |

## A.8 LIMITATIONS AND FUTURE DIRECTIONS

While AutoAgent presents a promising step toward democratizing LLM agent development through natural language alone, we identify several limitations that open compelling avenues for future exploration:

**Lack of Systematic Evaluation for Natural Language Agent Programming.** Although AutoAgent demonstrates strong capabilities in zero-code agent construction and self-customization, it currently lacks a systematic evaluation framework tailored to this novel paradigm. Existing benchmarks often focus on task execution performance, but do not adequately capture the effectiveness, reliability, and user experience of natural language-based agent programming. Future work could explore the development of a standardized evaluation suite that measures usability, correctness, adaptability, and end-user satisfaction in natural language-driven agent creation scenarios.

**Absence of GUI-based Agent Support.** While AutoAgent effectively supports tool invocation and DOM-level interactions in web environments, it does not currently incorporate GUI-based agent

capabilities. This limits its robustness in handling visually complex or dynamic web applications where visual cues and spatial layout play an essential role. Incorporating GUI agents—or hybrid agents that combine DOM and visual understanding—could enhance agent generalization and resilience, particularly in real-world human-computer interaction settings.

## B LLM USAGE

In the writing of this paper, the LLM was used only for correcting word choice and grammar. The LLM was employed exclusively as a research subject in the development of agents and was not involved in any critical stages of human-driven research, including idea generation, data analysis, or other core research activities.

