# OpenReview forum: "AutoAgent: A Fully-Automated and Zero-Code Framework for LLM Agents"
_ICLR.cc/2026/Conference — ICLR 2026 Conference Withdrawn Submission_

### Official Review · Reviewer_UKmv · 2025-10-28

**Soundness:** 3
**Presentation:** 3
**Contribution:** 3
**Rating:** 6
**Confidence:** 3

**Summary:**

This paper introduces AutoAgent, a novel framework designed to democratize the development of Large Language Model (LLM) agents by enabling fully automated, zero-code creation and customization through natural language. The core innovation lies in its modular architecture, which integrates four key components: Agentic System Utilities (for multi-agent orchestration), an LLM-powered Actionable Engine (for reasoning and tool use), a Self-Managing File System (for dynamic data handling), and a Self-Play Agent Customization module (for automated agent and workflow generation). The framework aims to bridge the accessibility gap in LLM agent development, which currently requires extensive programming expertise, by allowing users—including non-technical individuals—to build sophisticated agents via intuitive dialogue. The authors validate AutoAgent through comprehensive evaluations, including state-of-the-art performance on the GAIA benchmark (ranking #1 among open-source solutions) and superior results in Retrieval-Augmented Generation (RAG) tasks.

**Strengths:**

1. AutoAgent addresses a critical gap in LLM agent frameworks by eliminating the need for coding skills, making agent technology accessible to a broader audience. The natural language-driven approach—from agent creation to tool integration—represents a significant step toward inclusive AI development.
2. The framework's decomposition into four core components enables flexibility and scalability. The Agentic System Utilities facilitate seamless collaboration between specialized agents (e.g., Web, Coding, and File Agents), while the LLM-powered Actionable Engine supports both direct and transformed tool-use paradigms, ensuring compatibility with diverse LLM providers.
3. The paper provides strong experimental evidence across multiple benchmarks. AutoAgent achieves top-tier performance on GAIA (outperforming existing open-source methods) and excels in RAG tasks, highlighting its robustness.

**Weaknesses:**

1. The framework relies heavily on external LLMs (e.g., GPT-4, Claude) for core reasoning, which may introduce costs, latency, and reliability issues. The paper does not explore the impact of LLM quality variations or fallback mechanisms for open-source models, potentially affecting consistency in production use.

Missing reference:
Junyu Luo et al., Large language model agent: A survey on methodology, applications and challenges, arXiv 2025.

**Questions:**

Please see the weaknesses.

---

### Official Review · Reviewer_ndDo · 2025-10-31

**Soundness:** 2
**Presentation:** 2
**Contribution:** 2
**Rating:** 2
**Confidence:** 3

**Summary:**

The paper presents AutoAgent, a fully automated and zero-code framework for building LLM-based agents using natural language alone. In particular, it aims to democratize agent development by allowing non-technical users to create workflows through conversational interfaces. The proposed framework consists of four parts: (1) Agentic System Utilities, (2) a LLM-powered Actionable Engine; (3) a Self-Managing File System, and (4) a Self-Play Agent Customization module. The method achieves strong performance on the GAIA and Multihop RAG benchmark.

**Strengths:**

1. The paper is well motivated. It aims to address an important accessibility issue in LLM agent development by removing the coding barrier for non-technical users.
2. The proposed framework integrates multiple functional modules into a coherent and extensible agentic operating system.
3. The method achieves competitive results on GAIA and state-of-the-art RAG scores, demonstrating its effectiveness.

**Weaknesses:**

The paper's motivation is to make agent creation accessible to laypeople. However, this is not reflected in the evaluation. The experiments are all on standard benchmarks, without any realistic or interactive evaluation. Also, there is no user-centered experiment or study showing the usability for non-technical users. Besides, in my opinion, the paper sometimes overstates its contributions, e.g., calling it “revolutionary” when much of the technical content builds upon established frameworks.

**Questions:**

See weaknesses.

---

### Official Review · Reviewer_5DBV · 2025-10-31

**Soundness:** 3
**Presentation:** 3
**Contribution:** 2
**Rating:** 4
**Confidence:** 5

**Summary:**

AutoAgent introduces a fully automated, zero-code “agent OS” that lets users specify and deploy LLM agents, tools, and event-driven workflows entirely via natural language. The system combines (i) an orchestrator with Web/Coding/Local-File agents, (ii) an actionable engine that supports both native tool-use and XML-style transformed calls for reliability and model flexibility, (iii) a self-managing file system that ingests heterogeneous documents into a vector-DB for unified RAG, and (iv) a self-play customization pipeline that profiles requirements and auto-generates, tests, and refines tools/agents/workflows. Empirically, AutoAgent attains 55.15% average success on GAIA validation, leading open-source systems and ranking overall second behind a closed baseline, outperforming chunk-, graph-, and agentic-RAG baselines.

**Strengths:**

- The paper reframes agent development as zero-code, language-first engineering, combining native tool-use with XML-style calls and event-driven workflows in a novel, practical way.
- It delivers a coherent end-to-end “agent OS” with executable rigor (tool auto-testing, coding sandbox) and strong results on GAIA and MultiHop-RAG.
- The architecture is cleanly modularized with explicit interfaces and traceable, step-by-step creation logs that make reproduction straightforward.
- By removing coding barriers and offering reusable patterns, it meaningfully broadens access to agentic systems and is likely to influence future frameworks and evaluations.

**Weaknesses:**

- The claim of being the “first” or uniquely natural-language–driven is not well substantiated, since many agent frameworks are already prompt- or language-driven.
- The paper reads more like a carefully engineered stack than a conceptually new algorithm.
- The chosen name is easily confusable with existing systems claiming automatic agent generation[1], while this work still relies on a library of predefined agents/tools and a fixed orchestration backbone.
- Despite zero-code claims, the approach hinges on three built-in specialist agents (Web/Coding/File) and curated tool templates, which limits generality if the user’s domain requires new primitives.
- It is unclear how much each design choice contributes (dual tool-calling, event-driven workflows, vector-DB file system, handoff orchestration).
- The figure contains names, which may violate the double-blind review policy.

[1] AutoAgents: A Framework for Automatic Agent Generation, IJCAI, 2024.

**Questions:**

- How does your system detect or recover from tool/API failures and could you provide experiments where such failures are stressed?
- Can you report the cost and latency (token usage, wall-clock time, tool-call overhead) for your main tasks, and discuss how performance degrades under tighter budget constraints?
- Have you attempted constructing a brand-new specialist agent (outside Web/Coding/File) entirely from natural language instructions to demonstrate generalizability beyond built-in agent types?

---

### Note · Authors · 2026-01-06

**Comment:**

I want to withdrawal to revision.

**Withdrawal Confirmation:**

I have read and agree with the venue's withdrawal policy on behalf of myself and my co-authors.